# Predictive prioritization of enhancers associated with pancreatic disease risk

## Graphical abstract

## Highlights

- 3D enhancer-promoter contacts in five primary human pancreatic cell types

- Graph "tree" models reveal enhancer connectivity controlling cell identity

- Algorithm based on tree models identifies enhancers most critical for gene expression

- Predicted key enhancers are enriched for germline pancreatic disease risk variants

## Authors

Li Wang, Songjoon Baek, Gauri Prasad, ...,
Jason W. Hoskins,
Laufey T. Amundadottir, H. Efsun Arda

## Correspondence

efsun.arda@nih.gov

## In brief

Identifying key enhancers underlying disease susceptibility is a significant challenge. By mapping chromatin contacts between enhancers and promoters across five pancreatic cell types, Wang et al. built graph "tree" models capturing multi-enhancer control of genes. A ranking algorithm identified enhancers with the strongest influence on cell-type-specific expression, and perturbations in primary cells supported the predictions. The enhancers prioritized solely by this framework were enriched for risk variants, including those for diabetes and pancreatic cancer. The resulting maps and models provide a resource for variant-to-function studies in the human pancreas.

 Wang et al., 2026, Cell Genomics 6, 101040
January 14, 2026 Published by Elsevier Inc.

CellPress

## Article

# Predictive prioritization of enhancers associated with pancreatic disease risk

Li Wang,[1,4] Songjoon Baek,[1] Gauri Prasad,[1,2] John Wildenthal,[1] Konnie Guo,[1] David Sturgill,[1] Thucnhi Truongvo,[1] Erin Char,[2] Gianluca Pegoraro,[1] Katherine McKinnon,[3] The Pancreatic Cancer Cohort Consortium, The Pancreatic Cancer Case-Control Consortium, Jason W. Hoskins,[2] Laufey T. Amundadottir,[2] and H. Efsun Arda[1,5,*]

[1]Laboratory of Receptor Biology and Gene Expression, Center for Cancer Research, National Cancer Institute, National Institutes of Health, Bethesda, MD 20892, USA
[2]Laboratory of Translational Genomics, Division of Cancer Epidemiology & Genetics, National Cancer Institute, National Institutes of Health, Bethesda, MD 20892, USA
[3]Vaccine Branch, National Cancer Institute, National Institutes of Health, Bethesda, MD 20892, USA
[4]Present address: Department of Biology, Johns Hopkins University, Baltimore, MD 21218, USA
[5]Lead contact
*Correspondence: efsun.arda@nih.gov

## SUMMARY

Genetic and epigenetic variation in enhancers is associated with disease susceptibility; however, linking enhancers to target genes and predicting enhancer dysfunction remain challenging. We mapped enhancer-promoter interactions in human pancreas using 3D chromatin assays across 28 donors and five cell types. Using a network approach, we parsed these interactions into enhancer-promoter tree models, enabling quantitative, genome-wide analysis of enhancer connectivity. A machine learning algorithm built on these trees estimated enhancer contributions to cell-type-specific gene expression. To test predictions, we perturbed enhancers in primary human pancreas cells with CRISPR interference and quantified effects at single-cell resolution using RNA fluorescence *in situ* hybridization (FISH) and high-throughput imaging. Tree models also annotated germline risk variants linked to pancreatic disorders, connecting them to candidate target genes. For pancreatic ductal adenocarcinoma risk, acinar regulatory elements showed greater variant enrichment, challenging the ductal cell-of-origin view. Together, these datasets and models provide a resource for studying pancreatic disease genetics.

## INTRODUCTION

Pancreatic disorders, including diabetes mellitus, pancreatitis, and pancreatic cancer, impact over 10% of the global population, placing a significant burden on health and economic systems.[1–3] The pancreas, with its exocrine and endocrine compartments, plays vital roles in both digestion and glucose metabolism. These compartments arise from a common multipotent progenitor during embryonic development and are composed of distinct cell types, like α, β, δ, duct, and acinar cells.[4,5] Despite their specialized roles, pancreatic cells exhibit remarkable plasticity, with the potential for transdifferentiation and dedifferentiation.[6–8] While this phenotypic plasticity offers a regenerative potential for replacing lost or injured tissue, it also presents a vulnerability for developing malignancies.[9,10] For instance, in the case of pancreatic ductal adenocarcinoma (PDAC), growing evidence suggests that acinar cells also contribute to premalignant lesions by transdifferentiating into duct-like states through acinar-to-ductal metaplasia,[11–14] challenging the long-standing assumption that PDAC originates solely from ductal cells. Therefore, it is crucial to understand the underlying mechanisms that establish and maintain pancreatic cell identities,[15–18] as these

mechanisms are likely relevant for regenerative and oncogenic processes.

Enhancers are noncoding DNA elements that regulate gene expression through chromatin interactions and are key regulators in the establishment and maintenance of cell identities. Together with transcription factors, enhancers have an indispensable role in orchestrating tissue-specific gene expression patterns during development, homeostasis, and disease states.[19–21] Importantly, over 90% of single-nucleotide polymorphisms (SNPs) at disease-associated risk loci identified through genome-wide association studies (GWASs) are noncoding, with more than 80% of these found in enhancer regions.[22] However, identifying enhancers, assigning them to their target genes, and determining the specific conditions or cell types in which genetic variants impact enhancer function all remain significant challenges.[23] This is further complicated by the fact that many enhancers do not activate the closest promoters, and some are located at large distances from their targets.[24] In the human pancreas, several studies cataloged candidate enhancer regions through open chromatin analysis and epigenetic marks.[25–33] A few recent studies have initiated efforts to link enhancers to target genes by profiling 3D chromatin interactions in human

pancreatic cells; however, either their scope was limited to analyzing whole islets without cell-type resolution[34–36] or they were constrained by their small sample size.[37,38]

To address these gaps, we generated cell-type-specific, enhancer-promoter interaction datasets using donor pancreas from a comprehensive cohort, spanning 28 donors and five cell types. Overcoming the limitations of the standard pairwise loop analysis, we employed a network approach to parse complex chromatin interactions into tree models, revealing connectivity patterns between enhancers and promoters critical for gene expression. The tree models enabled the development of a machine learning algorithm designed to predict the impact of enhancer perturbations on cell-type-specific gene expression, assigning an "effect size" to each enhancer. To validate our predictions and tackle the challenge of measuring cell-type-specific enhancer perturbation effects in solid organs like the pancreas, we adapted a high-throughput-imaging-based approach to quantify the outcome of enhancer perturbations in single cells from donor tissue. Thus, our study presents a resource for identifying and validating critical enhancers involved in cell-type-specific gene expression while offering a framework for interpreting the genetic basis of complex diseases.

## RESULTS

### Mapping cell-type-specific enhancer-promoter interactions using donor pancreas tissue

Enhancer activity is highly cell type specific. To obtain pure $\alpha$, $\beta$, $\delta$, acinar, and duct cell populations from donor pancreas, we refined previously published cell-sorting methods, achieving over 95% cell purity in all populations[29,39,40] (see STAR Methods and Figures 1A, S1A, and S1B). Our protocol was tailored to be compatible with both assay for transposase-accessible chromatin (ATAC)-seq[41,42] and HiChIP[43] assays, facilitating simultaneous chromatin accessibility and 3D chromatin interaction profiling from the same batch of purified cells. Comparing the abundance of key marker gene transcripts in sorted cell populations demonstrated the effectiveness of our cell isolation strategy (Figures S1A and S1B). On these purified cell types, we performed ATAC-seq and HiChIP using an H3K27ac antibody—a histone modification that marks enhancer and promoter elements.[44–48] These experiments yielded an extensive dataset of 37 ATAC-seq and 29 HiChIP libraries, with each cell type and assay having at least four biological replicates (Table S2). HiChIP experiments generated more than 5.5 billion reads to allow profiling chromatin interactions at high resolution (Table S2). Across cell types, we obtained, on average, 116,935 accessible regions and 80,947 loops per donor (Tables S2 and S3).

Principal-component and Pearson correlation analyses showed consistent clustering of cell types in our chromatin datasets (Figures S1C–S1F). Looking at genomic loci near hallmark genes specific to each lineage revealed loops exclusively associated with the relevant cell types—glucagon (*GCG*) in $\alpha$ cells, insulin (*INS*) in $\beta$ cells, somatostatin (*SST*) in $\delta$ cells, trypsinogen (*PRSS1*) in acinar cells, and carbonic anhydrase (*CA2*) in duct cells (Figures 1B–1F). Taken together, our refined cell purification coupled to HiChIP assays captured cell-type-specific 3D chromatin interactions in human pancreas cells.

### Parsing enhancer-promoter interactions using graph-based tree models

To gain insight into the 3D chromatin organization in distinct pancreatic cell types, we employed a graph-based approach to visualize and analyze HiChIP interactions. In contrast to commonly used visualization methods such as chromatin contact matrices or loop arc plots, "tree" graph models facilitate the discovery of hierarchical structures and the incorporation of other epigenomic data types.[49]

To build the enhancer-promoter tree models, we first generated a list of consensus loops, representing all loops detected in our combined cell-type-specific data (see STAR Methods and Figure S2A). We then transformed these chromatin interactions into tree models, where the nodes represent either the enhancer or promoter anchors and the edges represent chromatin loops detected in our HiChIP experiments. Each tree is defined by its root promoter and therefore can only contain one promoter node (Figures 2A and S2B; Tables S4 and S5). We began assessing the connectivity with promoters, designating them as the base level—zero ($P_0$). Any enhancer directly connected to a promoter was assigned to the next tier—level 1 ($E_1$). Enhancers that link to $E_1$ enhancers, but not directly to the promoters, were then categorized as level 2 ($E_2$). Similarly, we designated the loops based on their connectivity ($L_1$, $L_2$, $L_3$, and so forth). This step-by-step assignment continued until all enhancers looping to the promoter were complete, ensuring that each enhancer's level represents its connectivity to the promoter (Figure S2B; see STAR Methods for details).

Parsing the chromatin data into enhancer trees revealed that our HiChIP experiments overwhelmingly captured enhancer-promoter interactions (78%); enhancer interactions that did not involve a promoter (orphan enhancers) constituted less than 1% of the data (Figures S2B–S2D). Among the enhancer-promoter interactions, $E_1$ enhancers (73%) and $L_1$ connections (80%) were the most abundant, suggesting that most enhancers loop to their targets directly (Figure 2B). While categorizing the connections, we also noticed frequent interactions *between* enhancers (i.e., $L_2$ and $L_4$, ~22% of total edges; Figure S2D). However, most of these enhancers had a shorter connecting path to a promoter; therefore, we further simplified the enhancer trees by pruning these connections (Figure S2B). At the end, the indirect loops only constituted ~11% of all enhancer-promoter interactions.

Since HiChIP assays are based on proximity ligation, we wondered if there is a distance bias for capturing more $E_1$ (direct) vs. $E_2$ (indirect) enhancers. However, when we analyzed the linear distance between these enhancers and their promoters, we found that the $E_1$ enhancers were typically located at a greater genomic distance (median of 275,552 bp) than $E_2$ enhancers (median of 149,825 bp, Figure 2C). Further, we assessed the paired-end tag (PET) counts of $L_1$ (connecting to $E_1$) or $L_3$ loops (connecting to $E_2$) as a measure for chromatin interaction frequency and found that $L_1$ loops overall exhibited higher PET counts in every cell type, suggesting that $E_1$ enhancers likely form more stable loops with their target promoters than other enhancers in the tree, regardless of the distance (Figure S2E).

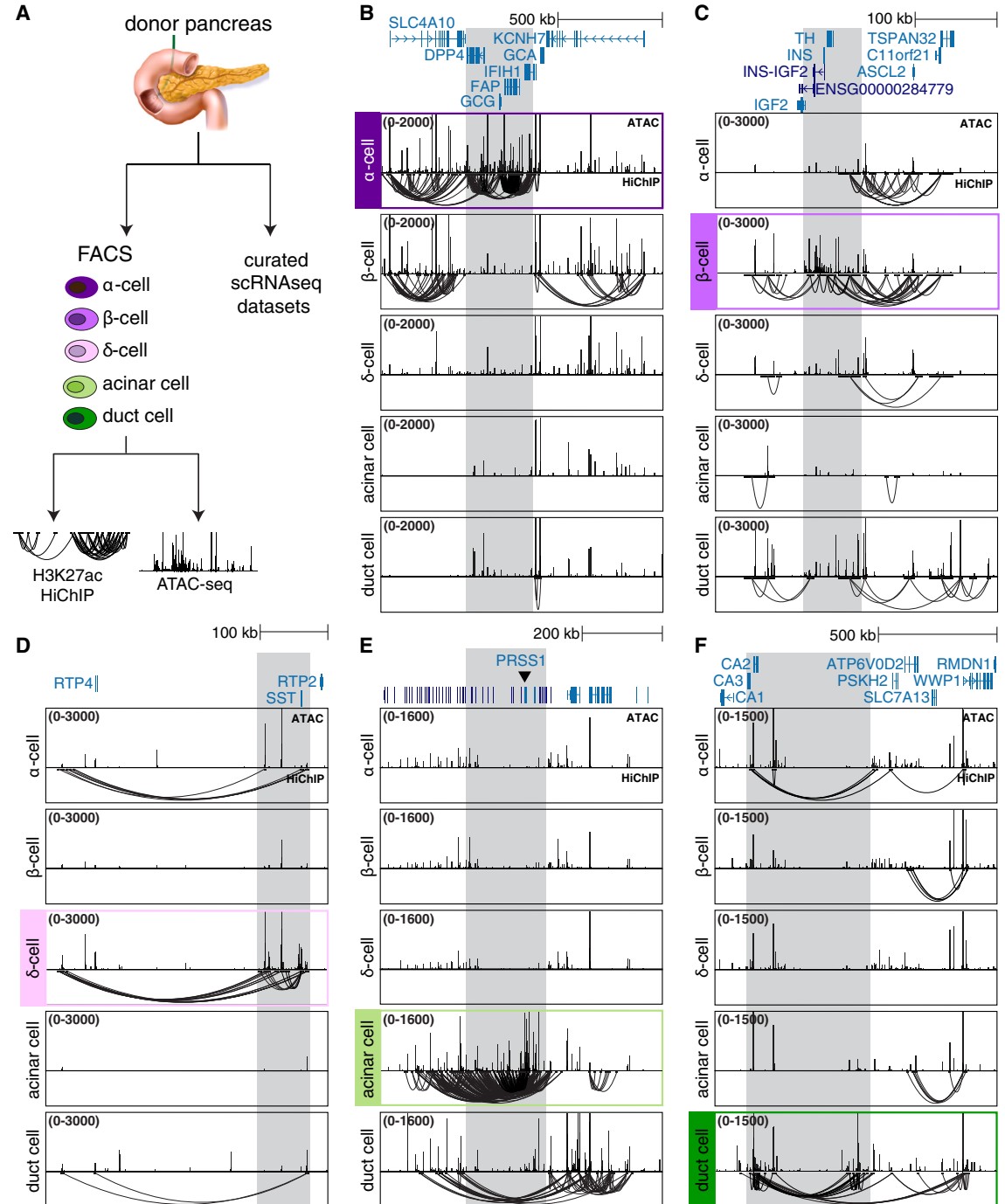

**Figure 1. Mapping cell-type-specific enhancer-promoter interactions using donor pancreas**

(A) Experimental overview.

(B–F) UCSC browser views showing ATAC-seq (top) and HiChIP (bottom) results in α, β, δ, acinar, and duct cells at hallmark loci highlighted in gray (*GCG*, *INS*, *SST*, *PRSS1*, and *CA2*). The gene models and scale are shown above each image.

The abundance of $E_1$ enhancers prompted us to investigate the functional impact of these direct interactions on target gene expression. We integrated our enhancer trees with gene expression data compiled from publicly available pancreas sin-gle-cell RNA sequencing (RNA-seq) experiments.[50] In all five pancreatic cell types, more than 80% of $E_1$ enhancers looped to a distal target promoter, bypassing genes closer in linear distance (Figure 2D). We found that the distally looped genes

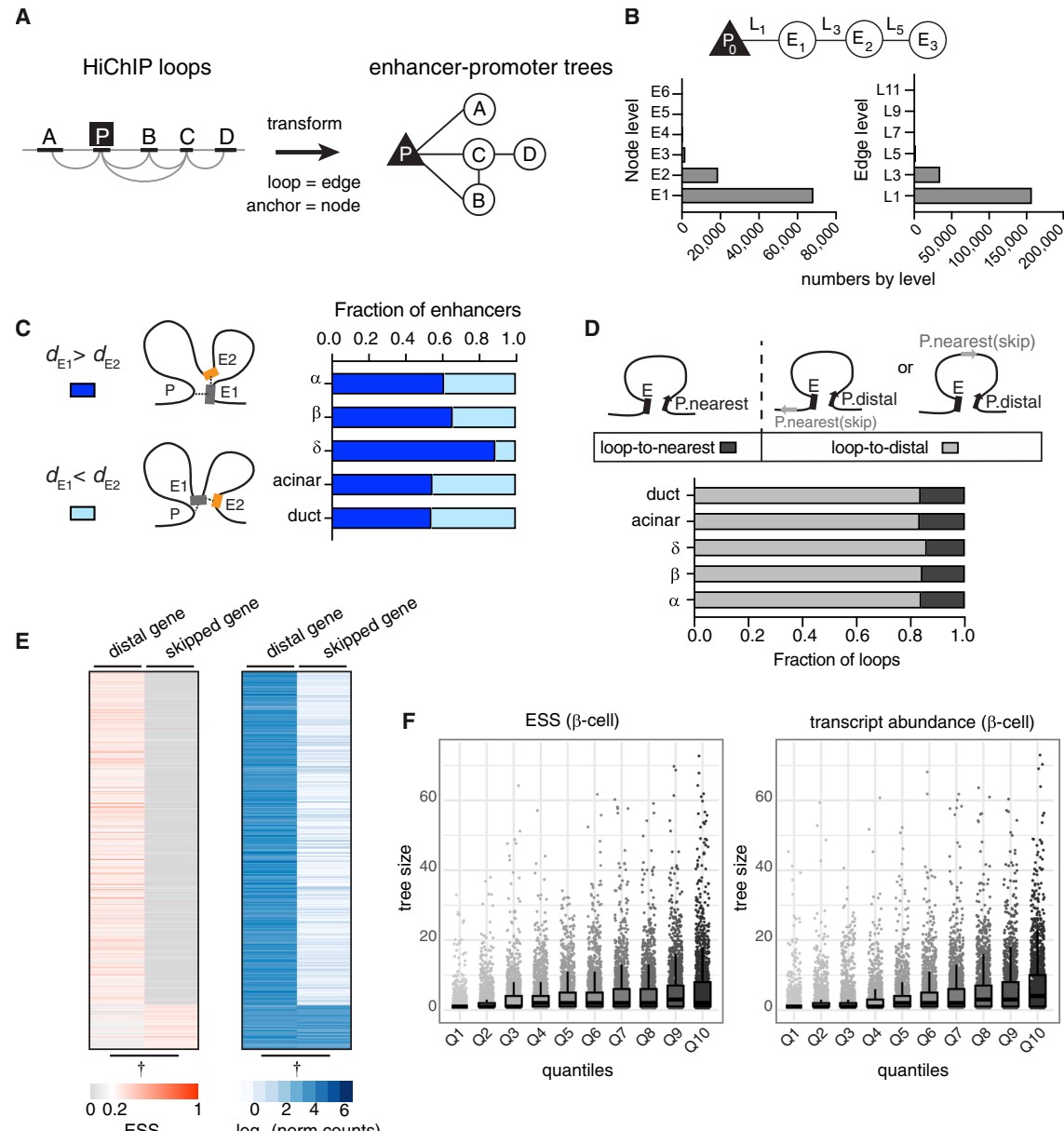

**Figure 2. Parsing enhancer-promoter interactions with tree models**

(A) Cartoon of HiChIP interactions transformed into tree models. The promoter forms the root, and enhancers become the branches.

(B) Enhancers (nodes) and loops (edges) are assigned levels based on their connectivity. Bar graphs show the distribution of node and edge numbers by level. See also Figure S2.

(C) Proportion of $E_1$ enhancers located further from the TSS than $E_2$ enhancers (dark blue) or closer (light blue). Possible looping configurations are shown.

(D) Fraction of enhancers looping to nearest (dark gray) or distal (light gray) genes by cell type.

(E) Heatmaps of expression specificity score (ESS; red) and abundance (blue) of distally looped or skipped genes in β cells. See Figure S2 for other cell types. Paired $t$ test, †$p < 0.001$.

(F) Boxplots depict the relationship between cell-type-specific gene expression (ESS), transcript abundance, and enhancer tree size in β cells. Whiskers extend to non-outlier points, and all data points are overlaid. Tree size is the number of enhancers linked to a promoter. See Figure S2I for other cell types.

have higher expression specificity and transcript abundance compared to skipped genes (Figures 2E and S3A). In addition, more than 68% of skipped genes were annotated as noncoding (Figure S3B). This trend persisted even when we limited our analysis to coding genes, with over 60% of $E_1$ enhancers still skipping the nearest gene (Figure S3C; Table S6).

We also examined the relationship between enhancer connectivity (tree size) and gene expression. Dividing the expression

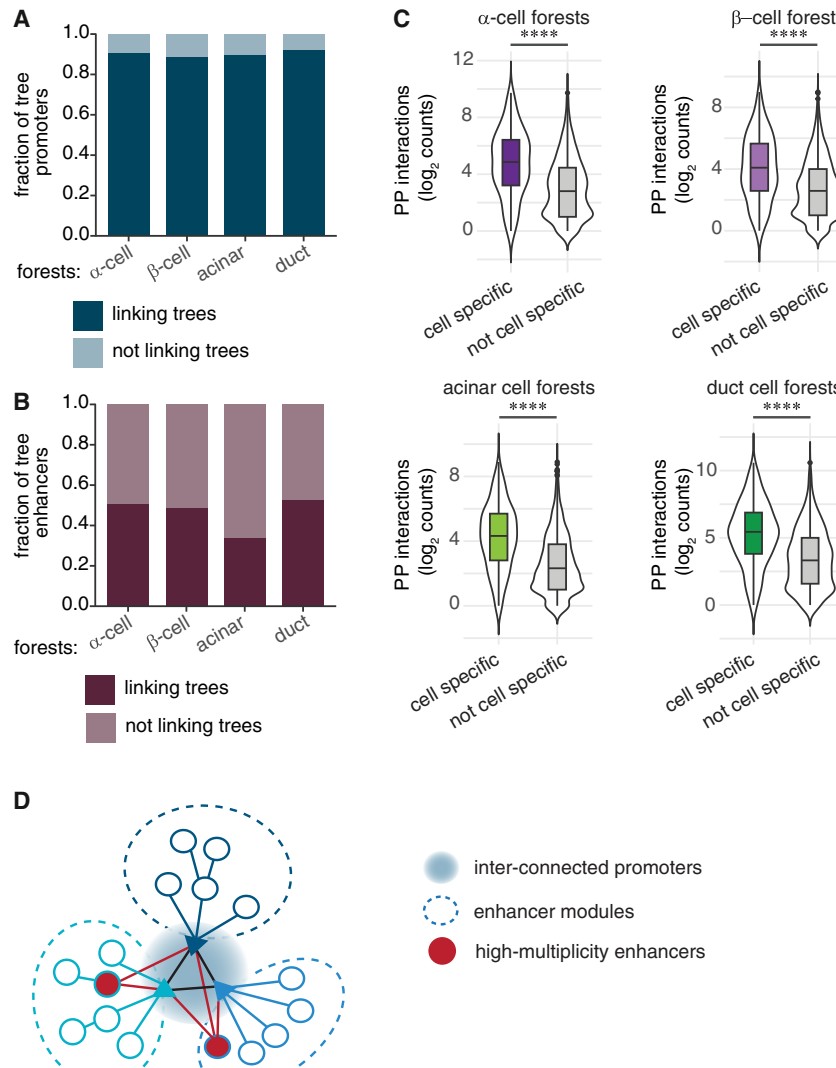

**Figure 3. Enhancer interconnectivity using tree models**

(A) Fraction of trees whose promoters connect to promoters of other trees in a forest.

(B) Fraction of trees with enhancers linking to other trees in a forest.

(C) Violin plots of promoter-promoter (PP) interactions in forests containing cell-type-specific promoters or not; embedded boxplots show the median and interquartile range, and whiskers extend to the most extreme data points that are not considered outliers. Mann-Whitney test, ****$p < 0.0001$.

(D) Model of enhancer forest structure. Promoters form a core that links the trees; most enhancers remain confined to "modules," with a few high-multiplicity enhancers bridging modules.

include genome-wide enhancer tree interconnectivity in different pancreatic cell types.

First, we assessed the extent of enhancers engaged with one or more promoters in our data. Across all cell types analyzed, we observed that, on average, 60% of enhancers only interacted with a single promoter, 21% interacted with two, and 19% interacted with three or more promoters (Figure S4A). Notably, we found that even when an enhancer loops to multiple promoters, its connectivity level rarely changed (Figure S4B), suggesting a simpler architecture rather than complex, multi-level enhancer networks.

To further explore the enhancer interconnectivity, we focused on enhancer trees that are connected to each other through a shared enhancer or promoter node. We termed these larger structures "enhancer forests." For this in-depth anal-

data into quantiles revealed that genes connected to multiple $E_1$ enhancers exhibited higher expression specificity and higher transcript abundance in each cell type (Figures 2F and S3D). We speculate that the $E_1$ enhancers may collectively promote transcription by increasing the local concentration of lineage-specific transcription factors, leading to robust expression of genes critical for cell identity in each cell type.

### Quantifying enhancer interconnectivity using tree models

In the previous section, we considered the connectivity of individual enhancer-promoter interactions. Enhancers, however, can regulate more than one gene and interact with multiple distinct loci. Standard pairwise loop analysis can underrepresent higher-order chromatin contacts or multiway interactions that may exist between multiple enhancer clusters.[51] Tree models address this by preserving connectivity and dependency between nodes, permitting the discovery of interactions involving more than two regions. Thus, we extended our analysis to

ysis, we proceeded with α, β, acinar, and duct cell data, excluding δ cell data due to the substantially lower number of enhancer trees detected (Table S7; see STAR Methods). We found that nearly all enhancer trees belong to a forest—in each cell type, 92%–97% trees are in forests (Figure S4C). The median size of a forest includes seven enhancer trees, with a median of 22 nodes, averaged across cell types (Figure S4D). The median genomic span of the enhancer forests is 732 kb (Figure S4D).

Enhancer trees can form a forest through connections between promoter or enhancer nodes. Analyzing these shared nodes revealed a striking pattern: across all cell types, approximately 90% of promoters connect to another promoter within their forest, whereas only 34%–52% of enhancers link distinct enhancer trees (Figures 3A and 3B). Furthermore, enhancer forests containing cell-type-specific genes showed a substantial increase in promoter-promoter interactions with a median of 50 compared to 10 in forests without cell-type-specific gene promoters (Figure 3C). The increased connectivity between promoters may facilitate the co-regulation of cell-type-specific

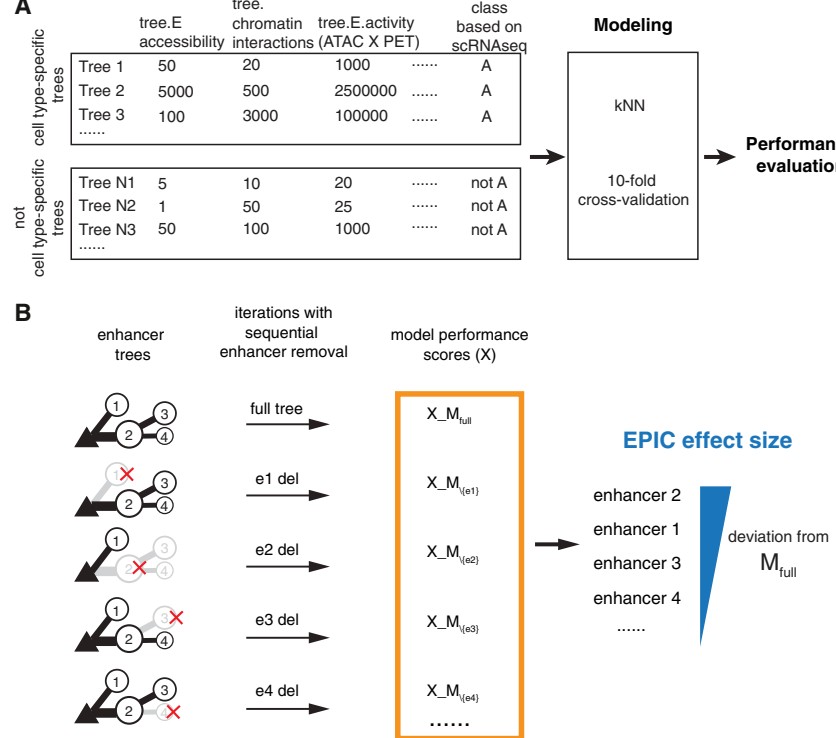

**Figure 4. EPIC: A machine learning model to prioritize enhancers influencing cell-type-specific gene expression**

(A) Schematic of the EPIC input features and classification approach. Enhancer trees were labeled as cell-type specific or not, based on single-cell RNA-seq data. Chromatin features trained a *k*-nearest-neighbor (kNN) classifier. Model accuracy was evaluated by 10-fold cross-validation.

(B) Enhancer prioritization by sequential removal of enhancer nodes from trees. EPIC effect size quantifies the impact on model accuracy; larger values mean there is a stronger influence on transcription.

from gene expression data obtained through single-cell RNA-seq studies in the human pancreas.[50] To assess the performance of EPIC, we evaluated additional models that included (1) only promoter accessibility, (2) enhancer association by linear genomic distance to transcriptional start site (TSS), and (3) a tree model with or without indirect features (Figure S5A). In all cell types examined, the tree models, which include the direct and indirect interactions, outperformed distance-associated or promoter-only models with the highest predictive accuracy (Figure S5B). There were only minor differences between the partial and full tree models, which is expected considering that indirect connections constitute only ~11% of the data (Figures 2B, S2C, and S2D).

Next, we asked if EPIC could predict the effect size of enhancer perturbations for a given target gene. Due to the inherent scalability of graph models, the enhancer trees can flexibly accommodate the addition or removal of nodes (enhancers) and edges (loops). Taking advantage of this feature, we systematically removed each enhancer node and compared the accuracy of the enhancer deletion models to that of the original model in predicting the correct class value—cell type (Figure 4B). Specifically, if an enhancer node was important for cell-type-specific expression, its removal would be expected to alter the probability score for that cell type, indicating a reduced confidence in the correct classification and potentially lowering the model's overall accuracy. Enhancers causing the most significant change in predicted probability were considered to have the highest effect size. This approach allowed us to evaluate, *in silico*, the effect size of perturbations for every enhancer in our enhancer trees.

### Testing EPIC's predictions in single cells using donor pancreas

A significant challenge in enhancer perturbation studies using primary human tissue is measuring perturbation effects in a cell-type-specific manner, especially in solid organs like the pancreas. To address this, we coupled RNA-fluorescence *in situ* hybridization (FISH) with high-throughput imaging[52,53] to dCas9-mediated gene activation (CRISPRa) or repression (CRISPRi)[54,55] and optimized the assays for donor pancreatic

genes to fulfill specialized functions in each cell type. Taken together, these results imply a more central role for promoters in forming the enhancer forests and suggest a modular topology of enhancers where each enhancer cluster typically regulates a specific set of connected genes (Figure 3D).

### Predictive prioritization of cell-type-specific enhancers using machine learning

In our investigation of the enhancer-promoter interactions within human pancreas cells, we identified a multitude of enhancers potentially contributing to gene regulation. However, it is unclear if all enhancers contribute equally to gene transcription or if some are more critical than others. We reasoned that our enhancer-promoter trees could facilitate functional prioritization of enhancers and, importantly, pinpoint those that may underlie disease risk. We developed a machine learning algorithm, EPIC (enhancer prioritizer using integrated chromatin data), capable of predicting the functional impact of enhancers on cell-type-specific gene expression. EPIC uses the *k*-nearest-neighbor algorithm and our enhancer trees to classify the cell-type specificity of these trees based on chromatin-derived features (Figure 4A). Specifically, we generated a list of 24 predictor variables (six variables per cell type) that include cell-type-specific chromatin accessibility (ATAC-seq tags), 3D chromatin interaction frequency (HiChIP PET counts), their interaction terms (ATAC × PET), and the enhancer tree structure (direct vs. indirect). EPIC uses these features to learn and make predictions about the cell-type specificity of the tree promoters (see STAR Methods for details; Figure 4A). The ground-truth class labels for the cell-type specificity of these tree promoters were derived

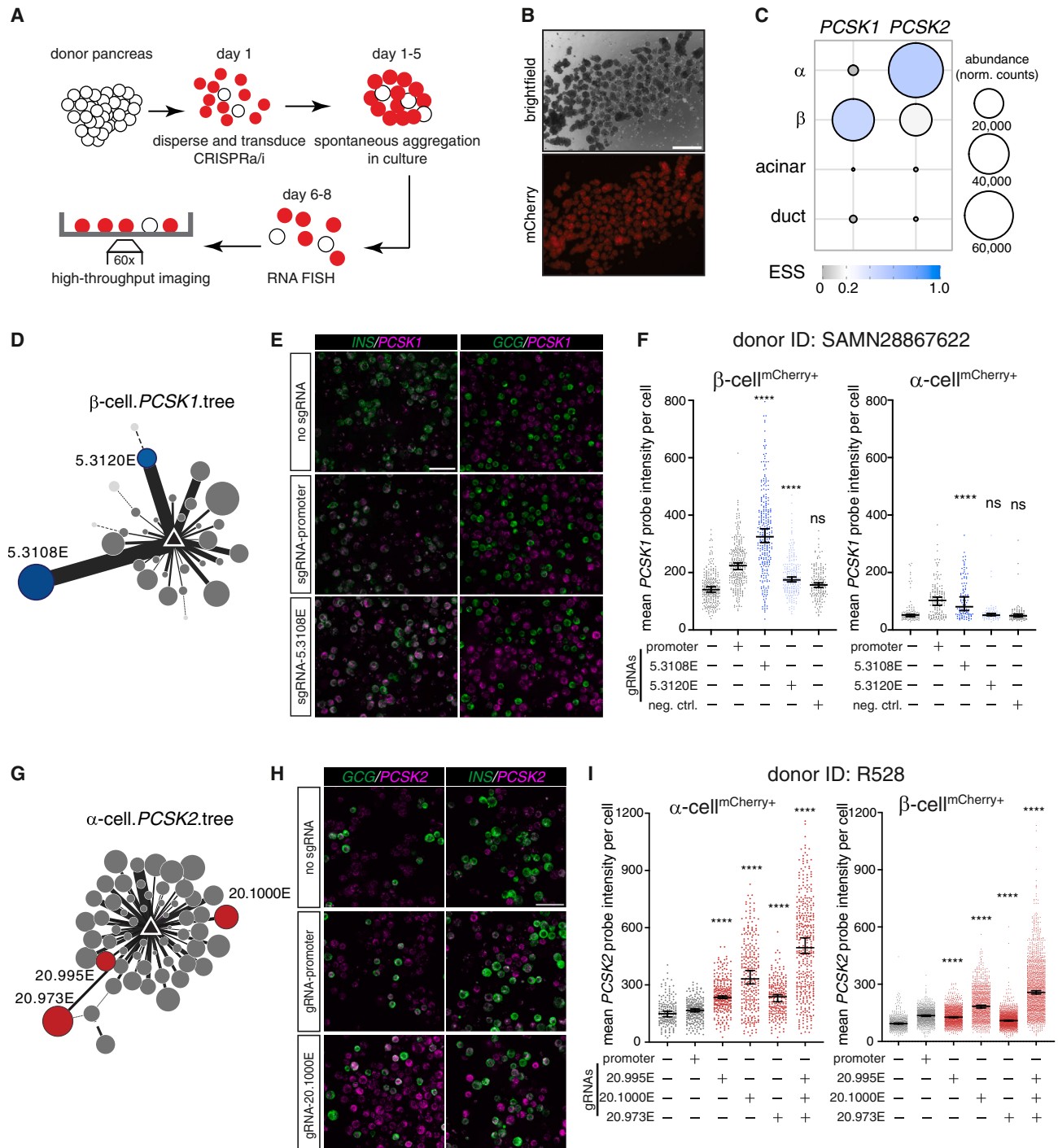

**Figure 5. Experimental validation of EPIC-prioritized enhancers in single cells using donor pancreas**

(A and B) Workflow schematic (A). Red cells represent dCas9-VP64/KRAB-mCherry+ cells after adenovirus transduction. Images in (B) show mCherry+ human pancreas cells after transduction. Scale bar: 500 μm.

(C) Bubble plots show ESS and abundance of *PCSK1* and *PCSK2* transcripts in human pancreas cells.

(D) β cell *PCSK1* enhancer tree. Node size represents ATAC-seq tag density, and edge width represents HiChIP interaction frequency in β cells. The blue-highlighted nodes indicate EPIC-prioritized enhancers. See also Figure S5.

(E) RNA FISH of *PCSK1* transcripts (magenta) in β (*INS* probe, green) and α (*GCG*, green) cells after CRISPRa. Scale bar: 50 μm.

*(legend continued on next page)*

cells (Figure 5A; see also STAR Methods). This approach ensured quantitative measurements of mRNAs at the single-cell level, in specific cell types, after each enhancer perturbation. Building on our prior success in delivering expression constructs to human pancreas cells,[29] we introduced adenoviral vectors carrying CRISPRa/i components and guide RNAs (gRNAs) targeting enhancers (see STAR Methods). On average, we achieved 60% transduction efficiency (Figures 5B and S6). After a 5-day recovery period in culture, we performed multiplexed RNA FISH to quantify the mRNAs of target genes and cell markers (Figure 5A; also see below).

To test EPIC's predictions, we focused on two loci—PCSK1 and PCSK2, due to their hallmark cell-type-specific expression (Figure 5C) and their critical roles in hormone processing.[56–60] Consistent with its role in converting proinsulin to its biologically active form, PCSK1 expression is most abundant in islet β cells (Figure 5C). In our HiChIP data, we detected over 30 enhancers forming loops to the PCSK1 promoter specifically in β cells, only seven in α cells, and none in exocrine cells (Figures S7A and S7B). Because there is no PCSK1 expression in α or exocrine cells, we opted to use the CRISPRa system (dCas9-VP64) to examine cell-type-specific effects of PCSK1 enhancer perturbations.[55] EPIC predicted 5.3108E and 5.3120E as top-ranking prioritized enhancers in β cells, with 5.3108E having a larger effect size (0.1) than 5.3120E (0.04) (Figures 5D and S7B; Table S8). We designed gRNAs targeting these two enhancer nodes (5.3108E located 32 kb and 5.3120E located 172 kb upstream of the TSS). We also designed additional gRNAs targeting the promoter as a positive control and a region 7.6 kb upstream of the TSS without any loops as a negative control (Figure S7A). After CRISPR targeting, we performed RNA FISH to quantify PCSK1 transcripts in specific pancreatic cells (Figure 5E). As expected, promoter-targeted dCas9-VP64 increased PCSK1 transcriptional output in β cells, while the negative control did not significantly alter PCSK1 levels (Figures 5F and S7C; Table S9). Furthermore, we found that modulating the enhancer regions with CRISPR activation resulted in increased PCSK1 transcription in the order that EPIC predicted for the effect sizes in β cells (Figure S7B; Table S8). Notably, we also observed upregulation of PCSK1 in α cells when the top-ranking enhancer, 5.3108E, was targeted (Figures 5F and S7C). In contrast, none of the targeted regions activated PCSK1 in exocrine cells (Figure S7D; Table S9).

PCSK2 is predominantly expressed in islet α cells, although single-cell RNA-seq studies demonstrated a PCSK2 transcript in β cells[50] (Figure 5C), and it has been implicated in the processing of proinsulin to insulin.[61–63] In line with this evidence, we observed 59 nodes in the PCSK2 enhancer tree in α cells, 24 in β cells, and zero in exocrine cells (Figures 5G and S8A). We designed gRNAs targeting three enhancers with effect sizes of 0.1

(20.973E), 0.07 (20.1000E), and 0.03 (20.995E) and the promoter as a positive control (Figures S8A and S8B; Table S8). Perturbing these three enhancers with CRISPRa resulted in increased PCSK2 transcription in α cells along with the promoter (Figures 5H, 5I, and S8C; Table S9). However, 20.1000E targeting caused the greatest increase, even though 20.973E had a slightly higher predicted effect size (Figures 5I and S8C). Looking into EPIC's effect size predictions for PCSK2 enhancers, we noticed that they had a narrow range (0.1 for the highest vs. 0.03 for the lowest; Table S8), potentially revealing regulatory redundancies and, consequently, the need to perturb more than one element at a time to observe significant alterations to transcription. Indeed, when we targeted all three enhancers simultaneously using CRISPRa, we observed a near-additive effect on PCSK2 transcript levels (Figures 5I and S8C). Perturbing these α cell-specific enhancer tree nodes in β cells also resulted in upregulation of the PCSK2 transcript (Figures 5I and S8C; Table S9) but did not do so in exocrine cells (Figure S8D), similar to the PCSK1 findings in α cells. These results suggest that the chromatin environment in α and β cells, but not in exocrine cells, is permissive to activation at these enhancers. This finding aligns with the fact that both α and β cells differentiate from the endocrine lineage during embryonic development.

Taken together, our results demonstrate the potential of our machine learning algorithm, derived from our enhancer tree data, to identify enhancers with a significant impact on target gene transcription.

## Enhancer trees and EPIC facilitate functional annotation of genetic variants associated with pancreatic diseases

Hundreds of genetic variants, including SNPs, have been linked to pancreatic diseases by GWASs.[32,64–66] However, how these variants impact the disease susceptibility remains largely unclear. Due to linkage disequilibrium (LD), disease-associated loci often contain multiple highly correlated SNPs, making it difficult to identify the causal mechanisms between variants and genes. In addition, determining which cell types are relevant to the disease is challenging since complex diseases, like diabetes or cancer, often involve interactions between multiple cell types. We reasoned that our cell-type-specific enhancer-promoter trees in the pancreas, combined with EPIC's ability to prioritize enhancers, could help annotate pancreatic-disease-associated variants by identifying candidate enhancers and their putative target genes.

First, we analyzed the enrichment of GWAS SNPs related to pancreatic disorders within enhancer or promoter nodes across pancreatic cell types using the GARFIELD algorithm[67] (see STAR Methods and Table S10). We found that type 2 diabetes (T2D) and glycemic trait variants were significantly overrepresented

(F) Quantification of PCSK1 in mCherry+ α cells and β cells. Each dot shows data from a single cell, and data are grouped by gRNA condition; lines indicate group medians, and error bars show 95% confidence intervals. One-way ANOVA with Dunnett's test. ****$p < 0.0001$ and ns, not significant. Additional donors are described in Figure S5 and Table S9.

(G) Same as (D) but for α cell PCSK2 enhancer tree. Red-highlighted nodes indicate EPIC-prioritized enhancers. See also Figure S5.

(H) Same as (E) but for PCSK2 transcripts. Scale bar: 50 μm.

(I) Quantification of PCSK2 in mCherry+ α cells and β cells. Dots represent single cell measurements by gRNA condition; lines show medians, and error bars show 95% confidence intervals. One-way ANOVA with Dunnett's test. ****$p < 0.0001$. Additional donors are described in Figure S5 and Table S9.

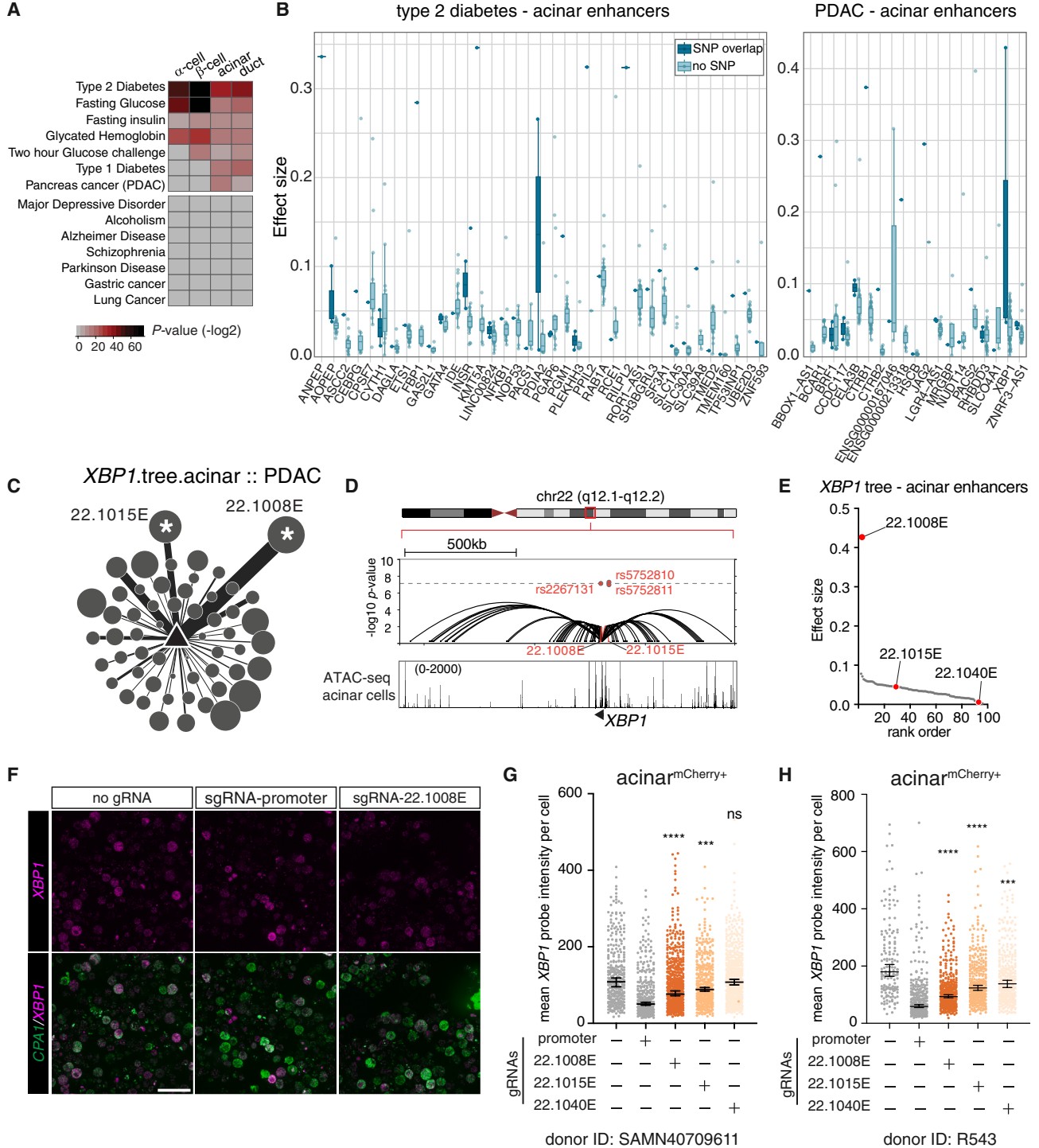

**Figure 6. EPIC prioritizes regulatory elements at pancreatic disease risk loci**

(A) Heatmap of risk SNP enrichment in cell-type-specific enhancer or promoter nodes.

(B) Distribution of effect sizes for acinar cell enhancers. Enhancers are grouped by gene, with dark blue points representing enhancers that overlap with a SNP and light blue points indicating those without an overlapping SNP. The whiskers extend to the most extreme data points that are not considered outliers.

(C) *XBP1* enhancer tree in acinar cells. Nodes size represents ATAC-seq tag density, and edge width represents HiChIP interaction frequency in acinar cells. * indicates SNP-enriched enhancers.

*(legend continued on next page)*

in islet cell enhancer trees (T2D in β cells: $p = 1.52 \times 10^{-23}$, GWAS threshold = $1 \times 10^{-7}$; fasting glucose in β cells: $p = 6.99 \times 10^{-23}$, GWAS threshold = $1 \times 10^{-5}$; and HbA1C in β cells: $p = 1.16 \times 10^{-10}$, GWAS threshold = $1 \times 10^{-5}$). In contrast, pancreatic cancer risk variants were significantly enriched within exocrine cell enhancer trees, with a more significant association observed in acinar cells ($p = 1.47 \times 10^{-5}$) compared to duct cells ($p = 3.65 \times 10^{-3}$, GWAS threshold = $1 \times 10^{-5}$) (Figure 6A). Non-pancreatic-disease traits, like Alzheimer's disease and lung cancer, showed no enrichment in pancreatic enhancer or promoter nodes, confirming the specificity of these associations (Figure 6A).

While the enrichment of T2D risk variants in islet cell enhancers is expected given their critical role in glucose metabolism, notably, we also observed significant enrichment of T2D risk variants in exocrine cell enhancers (e.g., in acinar cell enhancers, 141 total SNPs, 44 SNPs with GWAS $p < 1 \times 10^{-8}$, and 68 SNPs with $1 \times 10^{-8} < p < 1 \times 10^{-4}$). Closer examination of the tree promoters revealed genes with important functions in acinar cells (Figures 6B, S9A, and S9B; Table S10). For example, germline mutations in the transcription factor *GATA4* have been implicated in childhood-onset diabetes and exocrine deficiencies.[68] Our data placed T2D-risk SNPs in a *GATA4* enhancer at chr8p23.1, linking these variants to a putative gene target in acinar cells (Figure S9C).

To assess EPIC's ability to prioritize disease-relevant enhancers, we compared EPIC effect sizes of enhancers that overlapped pancreatic disease SNPs with those that did not (Table S10). Specifically, for a given cell type and trait association, we identified the SNP-enriched enhancer nodes, retrieved the enhancer trees that included those nodes, and ranked all the enhancers in the tree based on EPIC's effect size predictions (Figures 6B, S10A, and S10B). Using cumulative distribution plots, we found that SNP-overlapping enhancers ranked higher in EPIC effect sizes across three diseases (type 1 diabetes, T2D, and pancreatic cancer) and three cell types (β, acinar, and duct cells) (Figures S10C–S10H, Kolmogorov-Smirnov test, all $p < 1 \times 10^{-3}$). Thus, EPIC effectively prioritizes enhancers likely harboring regulatory variants associated with pancreatic diseases.

For PDAC, our analysis indicated a stronger enrichment of disease-associated variants in acinar cell enhancer trees, even though ductal cells historically have been the focus of PDAC research. We identified two acinar cell-specific enhancers at the *XBP1* chr22q12.1 locus enriched with three PDAC-risk SNPs (Figures 6C, 6D, and S11A). Of these two enhancers, one (22.1008E) overlaps with variant rs2267131 and has the highest EPIC effect size (0.43) for *XBP1* in acinar cells, while the other (22.1015E) has a lower effect size (0.05) and overlaps

variants rs5752810 and rs5752811 (Figures 6D and 6E). We targeted these enhancers using CRISPRi in acinar cells (Figure 6F), including an additional enhancer (22.1040E) that ranked among the lowest (Figure 6E and S11B–S11E; Table S8). We observed a significant reduction in *XBP1* transcripts in acinar cells targeting 22.1008E and 22.1015E, but not consistently in cells targeting the 22.1040E enhancer, demonstrating that the SNP-enriched 22.1008E and 22.1015E enhancers regulate *XBP1* transcription in acinar cells (Figures 6G and 6H). Importantly, the degree of reduction was in line with their EPIC-predicted effect sizes (Figure 6E).

Taken together, our chromatin-derived enhancer trees help annotate pancreatic-disease-risk SNPs as candidate functional variants influencing high-impact enhancers and link them to putative target genes in a cell-type-specific manner.

## DISCUSSION

Here, we presented a detailed map and analysis of enhancer-promoter interactions in primary human pancreas cells. By refining cell purification methods and coupling them to ATAC-seq and HiChIP assays, we have generated a high-resolution dataset that captures the enhancer-chromatin interactions, surpassing previous work in specificity, sample size, and depth.

Our analytical approach is based on versatile, graph-based tree models that simplify the representation and interpretation of chromatin interaction data, permitting systematic quantification of enhancer connectivity. As a result, our analysis demonstrated that direct enhancer ($E_1$) interactions are a predominant feature of gene regulation, with multiple enhancers collectively acting on individual promoters to increase transcript output and spatiotemporal specificity. This is consistent with prior studies in the developing human cortex[69] and mouse embryonic stem cell lineages,[70] where promoters with high levels of chromatin interactions correlated with lineage-specific genes and higher transcript levels. A similar principle was observed in a study[36] using human islets and promoter capture Hi-C to show dense 3D enhancer connectivity at genes critical for islet cell function. However, these prior studies primarily focused on the overall level of chromatin interactions without distinguishing the type of enhancer-promoter connectivity. Our tree models transformed these findings and found that most enhancers form direct loops to their target genes, supporting a simpler, potentially additive model for enhancing transcription output and specificity.[71,72] In addition, our forest analysis showed that extensive enhancer sharing between trees is not common, with most interconnectivity among trees occurring through promoter-promoter interactions. This suggests that enhancers might be forming distinct modules[73] with their respective

---

(D) Combined UCSC genome browser and locus zoom plots displaying the enhancer tree elements at the *XBP1* locus in acinar cells. (Top) GWAS *p* values for PDAC shown in red. (Bottom) ATAC-seq peaks and HiChIP loops detected in acinar cells. Red highlights the nodes enriched with the significant SNPs. See also Figure S8.

(E) Scatterplot of ranked effect size *XBP1* enhancers in acinar cells. CRISPRi-perturbed enhancers are in red.

(F) RNA FISH images showing *XBP1* (magenta) in acinar cells (*CPA1*, green) after CRISPRi perturbation. Scale bar: 50 μm.

(G and H) Quantification of *XBP1* mRNA in mCherry+ acinar cells after CRISPRi. Each dot shows data from a single cell, and data are grouped by gRNA condition; lines indicate group medians, and error bars show 95% confidence intervals. One-way ANOVA with Dunnett's test. ****$p < 0.0001$, ***$p < 0.001$, and ns, not significant. Results from two independent donors are shown. See Table S9 for additional data.

promoters within the nucleus (Figure 3D), potentially positioned for transcription activation when transcription factors reach sufficient local concentrations.[74]

Modifying or perturbing enhancer function is key to understanding enhancer dysregulation in disease.[23] Prior work on enhancer perturbation primarily used homogeneous cell lines, stem-cell-derived cells that have the advantage of unlimited expansion, or whole tissues without cell-type resolution.[33,36,75,76] Here, we used a high-throughput, single-cell imaging-based approach to measure the effects of enhancer perturbations in a structurally complex, solid organ like the pancreas.

Our chromatin data, combined with tree models, form the basis of EPIC, an algorithm designed to prioritize enhancers based on their predicted impact on cell-type-specific gene expression. We validated EPIC's predictions on a select number of enhancer trees and found that our orthogonal imaging-based enhancer perturbation results correspond well with the predicted importance. When analyzing the effect size distributions, however, we observed that the differences in EPIC-predicted effect sizes among enhancers within a given tree are relatively small. Furthermore, most enhancer trees contain only a few outlier enhancers, whose deletion causes significant deviation of EPIC prediction accuracy, indicating that such critical enhancers are rare. This scarcity of critical enhancers might reflect an evolutionary selection for robust lineage-specific gene expression, where multiple enhancers contribute additively or redundantly to ensure stable regulation. Consistently, our CRISPR-RNA FISH experiments demonstrated this additive effect on *PCSK2* enhancers in α cells. Further work involving the systematic perturbation of a combination of elements might help clarify the function of enhancers with smaller effect sizes.

Determining the cell type(s) affected by germline risk variants identified through GWASs and understanding their specific impact on gene regulation at scale remain ongoing challenges. Our cell-type-specific enhancer trees enabled us to nominate candidate functional variants and target genes at GWAS risk loci across several pancreatic diseases. While we confirmed previously known associations, we also identified novel links between risk SNPs and their putative transcriptional targets in understudied cell types.[77,78] Specifically, we linked acinar cells with the overall inherited risk of PDAC. We also observed substantial crosstalk between the exocrine and endocrine compartments of the pancreas in the context of enhancers and risk SNP associations, for instance, the enrichment of T1D and T2D SNPs within exocrine enhancers.[29,32] This crosstalk suggests a complex interplay between different cell types in disease susceptibility. More research is needed to understand how different types of cells contribute to the development and progression of pancreatic disorders.

Additionally, our analysis revealed that enhancers overlapping with risk SNPs are more likely to be top-ranking enhancers. Notably, in the trait-cell type associations we analyzed, we observed that some genes have top-ranking enhancers without SNP overlap. We speculate that these enhancers might harbor disease risk variants that are yet to be discovered. In a prior study, HiChIP assays facilitated the identification of promoter-interacting expression quantitative trait loci (pieQTLs) in immune cell types, supporting the view that chromatin features can reveal genetic variants that impact gene expression and disease susceptibility.[79]

While our analyses focused on common genetic variants associated with complex pancreatic traits, it is important to consider that rare, high-impact germline variants or mutations within enhancers similarly disrupt gene expression and lead to monogenic disorders. For instance, rare noncoding variants within an enhancer of the hexokinase gene (*HK1*) were recently shown to cause congenital hyperinsulinism by mis-regulating expression in pancreatic β cells.[80,81] Our enhancer maps and the EPIC prioritization framework could thus greatly facilitate the localization and mechanistic characterization of such variants, enabling deeper insights into the genetic basis of rare monogenic diseases.

Our study provides new tools and resources for prioritizing enhancers central to cell-type-specific gene expression. Exploring the associations we have discovered between genetic variants and their putative gene targets should advance our understanding of how enhancer dysfunction contributes to diseases, potentially accelerating the development of novel therapeutic strategies.

## Limitations of the study

A key limitation of Hi-C-based assays, including HiChIP, is the lack of single-cell resolution. These assays capture chromatin interactions as an average across potentially diverse chromatin conformations from millions of cells at a single point in time. This means that the interactions detected may reflect a composite view of different interaction subsets rather than uniform patterns across all cells. Therefore, it remains unclear whether the observed interactions represent persistent enhancer-promoter interactions in every cell or a sum of varying subsets present in different cells. Emerging single-cell technologies[82,83] may help clarify these distinctions and reveal dynamic enhancer usage based on transcriptional states within the same cell type.

## RESOURCE AVAILABILITY

### Lead contact

Requests for further information, resources, and reagents should be directed to and will be fulfilled by the lead contact, H. Efsun Arda (efsun.arda@nih.gov).

### Materials availability

CARGO array plasmids generated in this study will be available from the lead contact upon request.

### Data and code availability

- Genomic data have been deposited at GEO under SuperSeries accession number GEO: GSE245484 and are publicly available as of the date of publication. PDAC GWAS data have been deposited at dbGaP under accession numbers dbGaP: phs000206.v6.p3 and phs000648.v1.p1 and are publicly available as of the date of publication.
- All original code for enhancer network/tree generation and EPIC has been deposited at https://github.com/CBIIT/Arda-lab-CCR-NCI/tree/main/Wang_et_al_2025_EPIC_Scripts and archived at Zenodo (https://doi.org/10.5281/zenodo.17039381) and is publicly available as of the date of publication.
- Any additional information required to reanalyze the data reported in this paper is available from the lead contact upon request.

## CellPress

## CONSORTIA

The members of The Pancreatic Cancer Cohort Consortium are Jun Zhong, Demetrius Albanes, Gabriella Andreotti, Alan A. Arslan, Laura Beane-Freeman, Sonja I. Berndt, Julie E. Buring, Daniele Campa, Federico Canzian, Stephen J. Chanock, Yu Chen, Sandra M. Colorado-Yohar, A. Heather Eliassen, J. Michael Gaziano, Graham G. Giles, Phyllis J. Goodman, Christopher A. Haiman, Mattias Johansson, Verena Katzke, Charles Kooperberg, Peter Kraft, Manolis Kogevinas, I-Min Lee, Loic LeMarchand, Núria Malats, Satu Männistö, Marjorie L. McCullough, Roger Milne, Stephen C. Moore, Lorelei Mucci, Salvatore Panico, Alpa V. Patel, Ulrike Peters, Miquel Porta, Francisco X. Real, Howard D. Sesso, Xiao-Ou Shu, Meir J. Stampfer, Geoffrey S. Tobias, Kala Visvanathan, Elisabete Weiderpass, Nicolas Wentzensen, Emily White, Chen Yuan, Wei Zheng, Jean Wactawski-Wende, Rachael Z. Stolzenberg-Solomon, Brian M. Wolpin, and Laufey T Amundadottir.

The members of The Pancreatic Cancer Case-Control Consortium are Samuel O. Antwi, Paige M. Bracci, Steven Gallinger, Michael Goggins, Manal Hassan, Elizabeth A. Holly, Rayjean J. Hung, Donghui Li, Núria Malats, Rachel E. Neale, Kari G. Rabe, Harvey A. Risch, Herbert Yu, and Alison P. Klein.

## ACKNOWLEDGMENTS

We thank Y. Dalal, G. Hager, T. Misteli, S. Oberdoerffer, D. Larson, P. Batista, P. Rocha, F. Collins, J. Smith, W. Cui, and the members of the Arda, Collins, and Amundadottir laboratories for their help and critical review of this manuscript. We thank the Center for Cancer Research, National Cancer Institute core facilities: Genomics Core (Bethesda, MD), Sequencing Facility (Frederick, MD), Flow Cytometry Core (Bethesda, MD), and High-Throughput Imaging Facility (Bethesda, MD). This work used the computational resources of the National Institutes of Health (NIH) High Performance Cluster (Biowulf, https://hpc.nih.gov). This work was supported, in part, by the Intramural Research Program of the NIH, National Cancer Institute, Center for Cancer Research (grant no. ZIA BC011798 to H.E.A.). The contributions of the NIH author(s) were made as part of their official duties as NIH federal employees, are in compliance with agency policy requirements, and are considered works of the United States government. However, the findings and conclusions presented in this paper are those of the authors and do not necessarily reflect the views of the NIH or the US Department of Health and Human Services. The American Cancer Society (ACS) funds the creation, maintenance, and updating of the Cancer Prevention Study II cohort. The authors express sincere appreciation to all Cancer Prevention Study II participants and to each member of the study and biospecimen management group. The authors would like to acknowledge the contribution to this study from central cancer registries supported through the Centers for Disease Control and Prevention's National Program of Cancer Registries and cancer registries supported by the National Cancer Institute's Surveillance Epidemiology and End Results Program. Where authors are identified as personnel of the International Agency for Research on Cancer/World Health Organization, the authors alone are responsible for the views expressed in this article, and they do not necessarily represent the decisions, policy, or views of the International Agency for Research on Cancer/World Health Organization. We acknowledge funding for the Women's Health Study (WHS) source of data: CA047988, CA182913, HL043851, HL080467, and HL099355. We acknowledge the WHI investigators listed here: https://www-whi-org.s3.us-west-2.amazonaws.com/wp-content/uploads/WHI-Investigator-Short-List.pdf. The WHI program is funded by the National Heart, Lung, and Blood Institute, NIH, US Department of Health and Human Services, through 75N92021D00001, 75N92021D00002, 75N92021D00003, 75N92021D00004, and 75N92021D00005. The Integrated Islet Distribution Program is supported by NIDDK, grant no. 2UC4DK098085. We gratefully acknowledge the organ donors and their families.

## AUTHOR CONTRIBUTIONS

Conceptualization, L.W. and H.E.A.; investigation, L.W., T.T., J.W., and K.G.; software, S.B.; formal analysis, L.W., J.W., K.G., D.S., S.B., G.P., K.M., E.C., J.W.H., and H.E.A.; supervision, L.W., H.E.A., J.W.H., and L.T.A.; writing –

original draft, L.W., S.B., and H.E.A.; writing – review & editing, L.W., T.T., J.W., K.G., D.S., S.B., G.P., E.C., J.W.H., L.T.A., and H.E.A.

## DECLARATION OF INTERESTS

The authors declare no competing interests.

## STAR★METHODS

Detailed methods are provided in the online version of this paper and include the following:

- KEY RESOURCES TABLE
- EXPERIMENTAL MODEL AND SUBJECT DETAILS
  - Human pancreas tissue procurement
- METHOD DETAILS
  - Flow cytometry
  - Cell purity verification by quantitative RT-PCR (qPCR)
  - ATAC-seq assays
  - ATAC-seq data analysis
  - H3K27ac HiChIP assays
  - HiChIP data analysis
  - Constructing the enhancer tree models
  - Analysis of enhancer-promoter distances and skipped nearest promoters
  - Evaluation of enhancer importance by *in silico* perturbation (EPIC)
  - Perturbing enhancers using CRISPR interference in human pancreas cells
  - RNA-FISH using hybridization chain reaction (HCR) in primary human pancreas cells
  - GWAS risk SNP enrichment analysis in cell-type-specific enhancer trees

## SUPPLEMENTAL INFORMATION

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

## Article

CellPress

## STAR★METHODS

### KEY RESOURCES TABLE

| REAGENT or RESOURCE | SOURCE | IDENTIFIER |
|---|---|---|
| **Antibodies** | | |
| INS (D3E7)-biotin | Abcam | RRID:AB_445799 |
| anti-H3K27ac | Abcam | RRID:AB_2118291 |
| GCG(U16-850)-PE | BD Biosciences | BD565860 |
| SST-Alexa Fluor 488 | BD Biosciences | BD566032 |
| HPi2-Dylight 650 | Novus | RRID:AB_1723990 |
| HPx1-Dylight 488 | Novus | RRID:AB_1723850 |
| CD133/1(AC141)-PE | Miltenyi Biotec | RRID:AB_244342 |
| CD133/2 (293C3)-PE | Miltenyi Biotec | RRID:AB_244346 |
| Rat IgG | Thomas Scientific | C840F01 |
| eFluor450 | Thermo Fisher Scientific | 65-0863-18 |
| Streptavidin-APC | Thermo Fisher Scientific | 17-4317-82 |
| **Bacterial and virus strains** | | |
| One Shot™ Stbl3™ Chemically Competent E. coli | Thermo Fisher Scientific | C737303 |
| **Biological samples** | | |
| Human pancreas tissue | Integrated Islet Distribution Network (IIDP) University of Alberta Islet Core (Canada) Prodo Labs | https://iidp.coh.org https://www.bcell.org/adi-isletcore.html https://prodolabs.com/ |
| **Chemicals, peptides, and recombinant proteins** | | |
| HyClone FBS | Cytiva | SH30070.03 |
| CMRL 1066 | Corning Incorporated | 99-603-CV |
| NEBNext HF master mix | NEB | M0541 |
| NEB buffer 2 | NEB | B7002 |
| dNTP | NEB | N0446S |
| DNA Polymerase I, Large (Klenow) Fragment | NEB | M0210 |
| T4 ligase | NEB | M0202S |
| Accutase | Sigma-Aldrich | A6964-100ML |
| BSA | Sigma-Aldrich | 126609 |
| DNase-I | Sigma-Aldrich | DN25-1G |
| Dispase | Sigma-Aldrich | 04942086001 |
| Digitonin | Sigma-Aldrich | 300410 |
| Glycine | Sigma-Aldrich | 50046-50G |
| LiCl | Sigma-Aldrich | L7026 |
| MboI | Sigma-Aldrich | R0147 |
| Nicotinamide | Sigma-Aldrich | 72340 |
| NP40 | Sigma-Aldrich | I8896 |
| Tween 20 | Sigma-Aldrich | P1379 |
| Triton X-100 | Sigma-Aldrich | T8787 |
| Dynabeads Protein A | Thermo Fisher Scientific | 10001D |
| EDTA | Thermo Fisher Scientific | 15575020 |
| Formaldehyde | Thermo Fisher Scientific | 28906 |
| GlutaMAX | Thermo Fisher Scientific | 35050061 |

*(Continued on next page)*

*Continued*

| REAGENT or RESOURCE | SOURCE | IDENTIFIER |
|---|---|---|
| PBS | Thermo Fisher Scientific | 10010023 |
| Penicillin/streptomycin (P/S) | Thermo Fisher Scientific | 15140122 |
| Permeabilization | Thermo Fisher Scientific | 08-8333-56 |
| Pluronic-68 (Gibco) | Thermo Fisher Scientific | 24040–032 |
| Proteinase K | Thermo Fisher Scientific | 25530049 |
| RPMI 1640 | Thermo Fisher Scientific | 61870036 |
| Riboblock | Thermo Fisher Scientific | EO0384 |
| SDS | Thermo Fisher Scientific | 15553027 |
| Streptavidin C-1 beads | Thermo Fisher Scientific | 65001 |
| SYBR Gold | Thermo Fisher Scientific | S11494 |
| SYBR Green PCR Master Mix | Thermo Fisher Scientific | 4368708 |
| Critical commercial assays | | |
| Tn5 | Illumina | 20034198 |
| Zymo Direct-zol RNA MiniPrep Kit | Zymo Research | R2051 |
| DNA clean & Concentrator-5 Zymo kit | Zymo Research | D4014 |
| SuperScript III Reverse Transcriptase kit | Thermo Fisher Scientific | 18080–044 |
| Deposited data | | |
| GWAS summary statistics (for traits associated with pancreas disorders and control traits) | Type 2 diabetes Knowledge portal; MAGIC consortium; GWAS-EBI catalog | https://t2d.hugeamp.org/; http://magicinvestigators.org/downloads/;https://www.ebi.ac.uk/gwas/ |
| Genomic data | This paper | GEO: GSE245484 |
| PDAC GWAS data | This paper; Klein et al.[64] | dbGaP: phs000206.v6.p3 and phs000648.v1.p |
| Oligonucleotides | | |
| *INS* (Hs00355773_m1) | Thermo Fisher Scientific | 4331182 |
| *GCG* (Hs00174967_m1) | Thermo Fisher Scientific | 4331182 |
| *SST* (Hs00356144_m1) | Thermo Fisher Scientific | 4331182 |
| *CPA1* (Hs00156992_m1) | Thermo Fisher Scientific | 4331182 |
| *ACTB* (Hs01060665_g1) | Thermo Fisher Scientific | 4331182 |
| *CHGA* (Hs00154441_m1) | Thermo Fisher Scientific | 4331182 |
| *KRT19* (Hs00761767_s1) | Thermo Fisher Scientific | 4331182 |
| Recombinant DNA | | |
| Ad-CMV-NLS-dCas9-VP64-mCherry | Vector Biolabs | 90001 |
| Ad-CMV-NLS-dCas9-KRAB-mCherry | Vector Biolabs | modified from SFFV-dCas9-KRAB-BFP (addgene46911) |
| CARGO gRNA arrays | This paper (Table S11); Gu et al.[84] | N/A |
| Software and algorithms | | |
| FACS Diva 8 | BD Biosciences | https://www.bdbiosciences.com/en-us/products/software/instrument-software/bd-facsdiva-software |
| FlowJo10 | BD Biosciences | https://www.bdbiosciences.com/en-us/products/software/flowjo-v10-software |
| Geneious Prime 2023.0.1 | Geneious | https://www.geneious.com |
| GraphPad Prism 10.2.3 | GraphPadSoftware | www.graphpad.com |
| Columbus 2.9.1 | PerkinElmer/Revvity | https://www.revvity.com/product/image-data-storage-and-analysis-system-columbus |

*Continued*

| REAGENT or RESOURCE | SOURCE | IDENTIFIER |
| --- | --- | --- |
| Signal Imaging Artist (SiMA) 1.2 | PerkinElmer/Revvity | https://www.revvity.com/product/signals-image-artist-sw-5-user-subscript-inf02358 |
| QuantStudio Design and Analysis1.5.2 | Thermo Fisher Scientific | https://www.thermofisher.com/us/en/home/technical-resources/software-downloads/quantstudio-3-5-real-time-pcr-systems.html |
| GARFIELD | Iotchkova et al.[67] | https://www.ebi.ac.uk/birney-srv/GARFIELD/ |
| EPIC (*E*nhancer *P*rioritizer using *I*ntegrated *C*hromatin data) | This paper | https://doi.org/10.5281/zenodo.17039381 |
| PEPATAC 0.8.3 | Smith et al.[85] | https://pepatac.databio.org/en/latest/ |
| BEDtools | Quinlan et al.[86] | https://code.google.com/archive/p/bedtools/ |
| Homer suite 4.11.1 | Heinz et al.[87] | http://homer.ucsd.edu/homer/introduction/install.html |
| DE-Seq | Anders et al.[88] | https://bioconductor.org/packages/release/bioc/html/DESeq2.html |
| HiC-Pro | Servant et al.[89] | https://nservant.github.io/HiC-Pro/ |
| hichipper | Lareau et al.[90] | http://aryee.mgh.harvard.edu/hichipper |
| FitHiChIP | Bhattacharyya et al.[91] | https://github.com/ay-lab/fithichip |
| Cytoscape | Shannon et al.[92] | https://cytoscape.org/ |
| LocusZoom | Boughton et al.[93] | https://my.locuszoom.org/ |
| UCSC Genome Browser | Perez et al.[94] | https://genome.ucsc.edu/ |

## EXPERIMENTAL MODEL AND SUBJECT DETAILS

### Human pancreas tissue procurement

Pancreas tissue were procured through the Integrated Islet Distribution Program (IIDP, RRID:SCR_014387) at City of Hope, the Alberta Diabetes Institute IsletCore at the University of Alberta in Edmonton (http://www.bcell.org/adi-isletcore.html) with the assistance of the Human Organ Procurement and Exchange (HOPE) program, Trillium Gift of Life Network (TGLN), and other Canadian organ procurement organizations, and from Prodo Laboratories (Aliso Viejo, CA, USA). All pancreas tissue were obtained from non-diabetic adult organ donors who were deceased due to acute trauma or anoxia. Donor characteristics are described in Table S1. All studies involving human pancreas tissue were conducted in accordance with National Institutes of Health Institutional Review Board guidelines, and with approval from the Human Research Ethics Board at the University of Alberta (Pro00013094). All donor families provided informed consent for the use of pancreas tissue in research.

## METHOD DETAILS

### Flow cytometry
#### *Preparing human pancreas cells for flow cytometry*

Pancreas tissue was shipped to the laboratory by overnight delivery and processed immediately upon receipt without additional culturing time. The tissue was dissociated into single cells following the protocol described in Arda et al.,[29] with modifications. The tissue samples were pelleted at 200 RCF for 5 min and washed once with cold PBS buffer (Thermo Fisher Scientific 10010023) containing 0.1% Pluronic-68 (Gibco 24040-032, PBS-Plu Buffer). The washed tissue pellet was gently resuspended in PBS-Plu Buffer containing 50 µg/mL DNase-I (Sigma-Aldrich DN25-1G) and incubated in a 37°C water bath with gentle agitation for 7–10 min. After pelleting, the supernatant was discarded, and the tissue pellet was washed once with cold PBS-Plu Buffer. Next, accutase treatment (Sigma-Aldrich A6964-100ML) was performed to further dissociate the bulk tissue. Islets were resuspended in prewarmed accutase solution at 1000 IEQ/mL concentration, and exocrine tissues at 100 µL (packed pellet)/mL, then incubated in a 37°C water bath for 6–8 min with gentle agitation. The accutase was neutralized by adding an equal volume of cold PBS-Plu Buffer. The digested tissues were pelleted and washed once with cold PBS-Plu Buffer. A second round of digestion was performed to achieve single-cell dissociation. The pellets were resuspended in prewarmed PBS-Plu Buffer containing 50 µg/mL DNase I and Dispase (Sigma-Aldrich 04942086001) (0.1 U/1000 IEQ for islets, or 1 U/mL for exocrine tissue), and incubated in a 37°C water bath for

6–8 min with gentle agitation. The enzymes were neutralized by adding an equal volume of cold FACS Buffer (PBS containing 2% FBS [HyClone, Cytiva], 50 mM EGTA pH 8.0), followed by an additional wash with cold FACS Buffer. Finally, the dissociated cells were passed through a 70 μm mesh (Fisher Scientific 08-771-2) to remove cell debris.

### Flow assisted cell sorting (FACS)

Dissociated single pancreas cells were resuspended in FACS buffer. Prior to staining with specific antibodies, the cells were incubated on ice for 20 min with rat IgG (Thomas Scientific C840F01) (1μL per million cells) to block nonspecific binding, and eFluor450 fixable viability dye (Thermo Fisher Scientific 65-0863-18) to label dead cells. Cells were then washed once with FACS buffer and twice with PBS-Plu buffer. Specific antibody staining procedures for exocrine cells and islet cells are as follows.

*Exocrine cells.* We used HPi2-Dylight 650 (Novus, NBP1-18946C) to label and exclude islet cells, HPx1-Dylight 488 (Novus, NBP1-18951G) to label acinar cells and CD133/1(AC141)-PE (Miltenyi Biotec 130-080-801), CD133/2 (293C3)-PE (Miltenyi Biotec 130-090-853) to label the duct cells. See Figure S1A for the gating strategy.

*Islet cells.* We modified existing intracellular staining protocols to make the fixation conditions compatible with downstream ATAC-seq and HiChIP experiments.[40] Prior to staining with specific antibodies, the cells were fixed in 1% formaldehyde-PBS/Plu buffer (Thermo Fisher Scientific 28906) at 1 million cell per 1 mL for 10 min at room temperature with rotation. Fixation was stopped by adding glycine solution (125 mM final concentration) for 5 min at room temperature. Fixed cells were permeabilized with $1\times$ Permeabilization Buffer diluted in PBS (Thermo Fisher Scientific, 08-8333-56). After permeabilization, cells were stained with INS (D3E7)-biotin (Abcam ab20756) and Streptavidin-APC (Thermo Fisher Scientific 17-4317-82) to label β-cells, GCG(U16-850)-PE (BD565860) to label α-cells, SST-Alexa Fluor 488 (BD566032) to label δ-cells.

All antibodies were used at 1:100 (v/v) concentration, except for HPx1 (2 μL per million cell), INS (1:50) and GCG (1:50) in FACS buffer. All antibody incubation steps were performed on ice for 30 min. Labeled cells were sorted on a BD FACS Aria III (BD Biosciences) using a 100 μm nozzle and FACS Diva 8 software, with appropriate area scaling and doublet removal. Gates were determined using fluorescence-minus one (FMO) controls. Sorted populations were collected into low retention tubes containing 100–300 μL cold sort buffer containing 5% BSA (Sigma-Aldrich 126609) in PBS. Cytometry data were analyzed and plotted using FlowJo v.10 (BD Life Sciences). The total number of cells collected for each sample is listed in Table S2.

### Cell purity verification by quantitative RT-PCR (qPCR)

Total RNA was extracted from approximately 5,000 FACS purified cells using the Zymo Direct-zol RNA MiniPrep Kit (R2051) following manufacturer's instructions. Entire total RNA was used to synthesize the cDNA using the Invitrogen SuperScript III Reverse Transcriptase kit (Thermo Fisher Scientific 18080-044). qPCR reactions were set up using TaqMan gene expression assays and SYBR Green PCR Master Mix (Thermo Fisher Scientific 4368708). The following TaqMan probes were used: *INS* (Hs00355773_m1), *GCG* (Hs00174967_m1), *SST* (Hs00356144_m1), *CPA1* (Hs00156992_m1), *ACTB* (Hs01060665_g1), *CHGA* (Hs00154441_m1), *KRT19* (Hs00761767_s1). Each sample was run in triplicate on a QuantStudio 5 Real-Time PCR machine (Thermo Fisher Scientific). Enrichment of cell marker mRNAs was calculated using the delta(deltaCt) method relative to the mRNA levels in presort cells using the QuantStudio Design and Analysis Software (v1.5.2, Applied Biosystems). To estimate the purity of sorted populations, we used the method described in Bramswig et al.[27]

### ATAC-seq assays

We followed the Omni-ATAC-seq protocol described in Corces et al.[41] 6,900-50,000 sorted cells were used for each assay. To isolate the nuclei, cells were resuspended in cold ATAC-Resuspension Buffer (RSB) containing 0.1% NP40 (Sigma-Aldrich I8896), 0.1% Tween 20 (Sigma-Aldrich P1379), and 0.01% Digitonin (Sigma-Aldrich 300410), and incubated on ice for 3 min. The lysis was washed out by addition of 1 mL of RSB containing 0.1% Tween 20 and mixed by inverting tubes. The nuclei are then pelleted at 500RCF for 10 min at 4°C. For transposition, each nuclei pellet was resuspended in 50 μL of transposition mix (25 μL $2\times$ TD buffer, 2.5 μL transposase (100 nM final), 16.5 μL PBS, 0.5 μL 1% digitonin, 0.5 μL 10% Tween 20, 5 μL water) and incubated at 37°C for 30 min in a thermomixer with 1000 rpm shaking. After transposition, reactions were stopped by adding EDTA (Thermo Fisher Scientific 15575020) to a final concentration of 40 mM. For endocrine samples, reverse crosslink buffer (50 mM Tris, 1 mM EDTA, 1% SDS, 0.2 M NaCl) containing 0.2 mg/mL Proteinase K (Thermo Fisher Scientific 25530049) (add fresh) was added in $5\times$ volume to each reaction and incubated at 65C overnight. Transposed DNA fragments were purified using Zymo DNA Clean and Concentrator-5 Kit (Zymo D4014) and pre-amplified for 5 cycles, for which each reaction contained 2.5 μL of 25 μM i5 primer, 2.5 μL of 25 μM i7 primer, 25 μL $2\times$ NEBNext master mix (NEB M0541), and 20 μL transposed/cleaned-up DNA. PCR conditions (72°C for 5 min, 98°C for 30 s, followed by 5 cycles of [ 98°C for 10 s, 63°C for 30 s, 72°C for 1 min] then hold at 4°C. To determine additional cycles for each sample, 5 μL of the pre-amplified mixture was used to run qPCR in 15 μL reaction containing 3.76 μL sterile water, 0.5 μL 25 μM i5 primer, 0.5 μL 25 μM i7 primer, 0.24 μL $25\times$ SYBR Gold (Thermo Fisher Scientific S11494 in DMSO), 5 μL $2\times$ NEBNext master mix, with the following cycling conditions, 98°C for 30 s, followed by 20 cycles of [98°C for 10 s, 63°C for 30 s, 72°C for 1 min]. Additional cycles were calculated following method by Buenrostro et al.[42] The remaining 45 μL of pre-amplified samples were then further amplified accordingly. The PCR condition was 98°C for 30 s, followed by n cycles of [98°C for 10 s, 63°C for 30 s, 72°C for 1 min] then hold at 4°C. Final PCR reactions were purified using the Zymo kit. Libraries were quantified on Agilent TapeStation 4000 and mixed in 1:1 molar ratio, then sequenced on Illumina NextSeq550 to obtain 75 bp paired-end reads.

### ATAC-seq data analysis

We used PEPATAC version 0.8.3[85] with default parameters to process the ATAC-seq FASTQ files. Specifically, reads were aligned to hg38 using Bowtie with the '-X 2000', mitochondrial and blacklisted regions were removed, peaks were called using MACS2 with '-f BED -q 0.01 –shift 0 –nomodel'. All samples included in this study surpassed the current ENCODE quality standards for ATAC-seq data, with TSS enrichment scores of 15-26, and deduplicated aligned reads greater than 14 million per sample. The PEPATAC output bigwig files were used to visualize peaks in the UCSC genome browser.

To generate a consensus peak file, the peak coordinates from the narrowPeaks output files were merged using BEDtools[86] with a merge gap of 0 bp. We used the annotatePeaks.pl tool in the HOMER suite[87] to generate a count matrix corresponding the peak regions. The peaks were then filtered by excluding those with less than the median ATAC-seq signal for each sample. The final counts table contained a total of 371,234 peaks from 37 ATAC-seq samples. To identify cell-type-specific accessible chromatin regions, we used DESeq package[88] and performed different cell type or group comparisons: compare between acinar and duct; compare between $\alpha$, $\beta$, and $\delta$; compare endocrine cells ($\alpha$, $\beta$, $\delta$) and exocrine cells (acinar, duct). Peaks that passed the significance threshold of FDR $\leq$ 1E−06 were selected, resulting in a total of 108,042 differentially accessible peaks. We assigned the cell-type-specific clusters to these peaks using k-means clustering with correlation as the similarity metric. The coordinates of ATAC-seq peaks and associated cell type clusters are listed in Table S3.

### H3K27ac HiChIP assays

To perform the HiChIP assays, we obtained on average 600,000 purified acinar, duct, $\beta$- or $\alpha$-cells, and 80,000 $\delta$-cells and due to their low abundance. HiChIP libraries were prepared following procedures described in Mumbach et al.[43] with modifications applicable for low cell input material. Briefly, if the cells were not already fixed prior to sorting, they were fixed in 1% formaldehyde at $1 \times 10^6$ cell/mL concentration for 10 min at room temperature with rotation, quenched with glycine at final concentration of 125 mM for 5 min, then pelleted at 500RCF for 5 min and washed once with PBS/Plu buffer.

#### In situ contact generation

Fixed cells were resuspended in 250 μL of ice-cold Hi-C Lysis Buffer and rotate at 4°C for 30 min. The nuclei were pelleted at 2500RCF for 5 min and washed with 250 μL of ice-cold Hi-C Lysis Buffer. The washed nuclei pellet was resuspended in 50 μL of 0.5% SDS (Thermo Fisher Scientific 15553027) and incubated at 62°C for 10 min, then quenched by adding 146 μL of water and 25 μL of 10% Triton X-100 (Sigma-Aldrich T8787) with incubation at 37°C for 15 min, rotating end-to-end. To digest the chromatin *in situ*, 100U MboI (Sigma-Aldrich R0147) and 25 μL 10xNEB buffer 2 (NEB B7002 RT) were added to the reaction and incubated at 37°C for 2 h with rotation. Enzymes were heat-inactivated at 62°C for 20 min. To label ends of the digested chromatin fragments, 14 μL of dNTP (NEB N0446S) (dATP-14-biotin [Axxora, NU-835-BIO14-S] and dTTP, dCTP, dGTP at a final concentration of 40 μM each) and 15U of DNA Polymerase I, Large (Klenow) Fragment (NEB, M0210) were added and incubated at 37°C for 45 min. Ends were ligated by adding 484 μL of ligation mixture containing 75 μL 10× T4 ligase buffer, 62.5 μL 10% Triton X-100, 3.75 μL 20 mg/mL BSA, 2000U of T4 ligase (NEB, M0202S) and 337.75 μL water and incubated at RT for 2 h with rotation. The ligation mix was pelleted at 2500RCF for 5 min at 4°C.

#### H3K27ac-ChIP

The in situ Hi-C pellet was lysed in 130 μL of Nuclei Lysis Buffer then transferred to an AFA microtube (Covaris 520045) and sonicated on a Covaris E220 under the following setting: PIP 105W, duty factor 2%, CPB 200, time 4 min, temperature 6°C. The sheared lysate was cleared by centrifugation at 16100RCF, 4°C for 15 min. Cleared lysate was diluted 2× in ChIP Dilution Buffer and precleared with 15 μL Dynabeads Protein A (Thermo Fisher Scientific 10001D) at 4°C for 1 h with rotation. 1 μL of anti-H3K27ac (Abcam ab4729 Lot# GR3211959-1) was added to the precleared lysate to immuno-precipitate (IP) H3K27ac associated, proximity ligated chromatin fragments overnight at 4°C with rotation. 15 μL Dynabeads Protein A were added and incubated at 4°C for 2 h to pull down the IP-ed complex. Beads were washed in order of the following: Low Salt, High Salt and LiCl (Sigma-Aldrich L7026) buffers. Washing was performed at room temperature on a magnet stand by adding to a sample tube 300 μL of a wash buffer, turning the tube 180° relative to the magnet for few times, allowing the beads to set for 2 min then removing the supernatant. ChIP DNA was eluted by incubation in 50 μL ChIP Elution Buffer on a thermomixer (Eppendorf) at 37°C with 300 rpm mixing. Two elution cycles were performed per sample and the eluates were pooled. To reverse crosslinks, 5 μL Proteinase K (Thermo Fisher Scientific 25530049) was added and the mixture was incubated at 55°C for 45 min followed by 67°C for 2.5 h. The DNA was purified by the Zymo kit (Zymo D4014) following the manufacturer's instructions, and eluted with 12 μL water, of which 2 μL was used for quantitation on Agilent TapeStation 4000.

#### Biotin pull-down and sequencing library preparation

5 μL Streptavidin C-1 beads (Thermo Fisher Scientific 65001) were washed with 300 μL Tween Wash Buffer, resuspended in 10 μL 2xBinding Buffer and mixed with each sample. The mixtures were incubated at RT for 15 min with rotation. Beads were washed in 300 μL Tween Wash Buffer twice at 55°C with shaking (400 rpm), followed by one wash in 100 μL 2× TD Buffer. For on bead tagmentation, beads were resuspended in 50 μL mixture containing 25 μL 2xTD buffer, 0.05 μL Tn5 (Illumina 20034198) per 1 ng post-ChIP DNA and water and incubated at 55°C with interval shaking for 10 min. The tagmented beads were incubated in 300 μL 50 mM EDTA at 50°C for 30 min, followed by washes in 2 × 50 mM EDTA (300 μL) at 50°C for 3 min, 3× Tween Wash Buffer (300 μL) at 55°C for 2 min then once in 200 μL 10 mM Tris. To prepare sequencing library, beads were resuspended in 50 μL PCR mix (25 μL 2xNEB HF master mix (NEB M054), 1 μL 12.5 μM Nextera ad-noMx, 1 μL barcoded 12.5 μM Nextera ad2.x and 23 μL water) and amplified by PCR program: 72°C for 5 min, 98°C for 1min, followed by n cycles of [98°C for 15 s, 63°C for 30 s, 72°C for 1 min] then hold at 4°C. N was

determined by the amount of post-ChIP DNA as described in Mumbach protocol.[43] The libraries were cleaned up using the Zymo kit and eluted in 15–18 μL water, 2 μL of which was used to determine the quantity and size distribution on an Agilent TapeStation 4000. High quality libraries were sequenced as 75 bp paired-end runs on Illumina NextSeq500 or HiSeq4000 instruments.

### HiChIP data analysis
Paired-end reads from 29 HiChIP samples were aligned to hg38 using the Hi-C Pro pipeline[89] with default settings. Hi-C Pro trims, aligns, assigns reads to MboI restriction fragments, filters for valid interactions and generates binned interaction matrices.

#### Calling loops
We used hichipper[90] and FitHiChIP[91] to call loops on the HiC-Pro processed samples. For hichipper, we used the peak calling options *EACH, SELF* which include each sample individually and only self-ligation reads. hichipper interactions were filtered based on a PET count $\geq 2$ and a Mango FDR $\leq 0.01$, which are classified as "significant loops". For FitHiChIP, we used the "loose" parameter and a bin size of 5000 bp to call loops at FDR $\leq 0.05$. Loop tables were converted to *bigBed* format then uploaded to UCSC browser for visualization. To evaluate variability across donor samples, PET counts from individual donor loop sets were used as input for the principal component analysis (PCA). To calculate and visualize the PCA of donor samples, the R prcomp() function and the ggplot2 package were used.

#### Generating consensus loops
To construct this loop set, we merged the FASTQ reads of the 29 HiChIP samples based on cell type (α-cell, β-cell, δ-cell, acinar, and duct). We processed the merged reads using the HiC-Pro pipeline, and performed loop calling using hichipper and FitHiChIP as described above. We then intersected the hichipper and FitHiChIP loops for each cell type, using the BEDTools pairToPair function, which reports the overlaps if two loops have the same anchors. We considered these common loops 'high confidence loops' as they were identified by two independent loop callers. To obtain a final list of non-overlapping anchors, we merged 1,144,820 anchors from high-confidence loops across five cell types using BEDTools with the default 'any overlap' option, resulting in 127,487 consensus anchors. For each consensus anchor, the midpoint was determined by calculating the average of the start and end positions of the merged anchors, rounded down to the nearest integer. A consensus loop table was also created, in which each row logs a unique consensus loop, and corresponding PETs in every cell type in columns. The length of consensus loop was defined as genomic distance between anchor midpoints. After removing loops with length less than 5 kb (artifact of anchor merging process), we obtained a total of 349,749 consensus loops from all five cell types.

The *AnnotatePeaks.pl* function of Homer suite (version 4.11.1)[87] was used with hg38 build (gencodev42 (GRCh38.p13)) to annotate anchors. We defined the promoter regions as 2 kb upstream and 3 kb downstream of the transcription start site (TSS). If the midpoint of an anchor falls into a promoter region, that anchor became a promoter anchor (P), otherwise, an enhancer anchor (E). Enhancer anchors were named using chromosome location, a numerical identifier, and the letter "E" (e.g., 22.1008E). Promoters followed a similar notation, with the letter "P" used instead.

### Constructing the enhancer tree models
Principal network elements were defined using consensus loops. Anchors were represented as nodes (Table S4) and the loops connecting the anchors were represented as network edges (Table S5).

#### Notations
$N$ represents the set of all nodes.
$E$ represents the set of all edges (loops).
$L_N(n)$ represents the level of node $n$.
$L_E(e)$ represents the level of edge $e$, where $e$ connects nodes $n_1$ and $n_2$.
For an edge $e$ connecting nodes $n_1$ and $n_2$, let $e = (n_1, n_2)$.

#### Assigning levels to nodes:
For each iteration until no new nodes are assigned:
For each edge $e = (n_1, n_2)$ with either $L_N(n_1) \neq -\infty$ or $L_N(n_1) \neq -\infty$

$$\begin{cases} L_N(n_1) = L_{N(n_2)} + 1, & \text{if } L_N(n_2) \neq -1 \text{ and } L_N(n_1) = -\infty \\ L_N(n_2) = L_N(n_1) + 1 & \text{if } L_N(n_1) \neq -1 \text{ and } L_N(n_2) = -\infty \end{cases}$$

#### Assigning levels to edges
For each edge $e = (n_1, n_2)$,

$$L_E(e) = \begin{cases} L_N(n_1) + L_N(n_2), & \text{if } L_N(n_1) \neq -\infty \text{ and } L_N(n_2) \neq -\infty \\ -1, & \text{otherwise} \end{cases}$$

Nodes are assigned levels based on their proximity to a starting node (nodeid). The starting node is assigned a level of 0. In each subsequent iteration, adjacent nodes are assigned a level that's one greater than the current node. The level of an edge (or loop) is determined by the sum of the levels of the nodes it connects. If either of the nodes has not been assigned a level, the edge remains unassigned (level = $-\infty$).

### Redundancy removal

Delete edges $e = (n_1, n_2)$ with $L_N(n_1) = L_N(n_2)$.

Consensus enhancer trees were built following these steps described above. Regarding cell-type-specific enhancer trees, throughout the manuscript, we use the term to describe enhancer-promoter interactions detected specifically within purified cell populations. This designation emphasizes the cell-type-resolved nature of the data and does not necessarily imply differential interactions. Consequently, enhancer trees defined for one cell type may contain (1) interactions unique to that cell type, (2) interactions shared among related cell types, or (3) interactions common across multiple pancreatic cell populations. For instance, α-cell trees can contain enhancers common across multiple cell types (like *ACTIN* locus enhancers), enhancers shared within compartments (like *CHGA* for endocrine), and enhancers specific to α-cells (like the *GCG* locus). We have obtained 15557 enhancer trees for α-cells, 14276 for β-cells, 1580 for δ-cells, 10949 for acinar cells, and 12483 for duct cells. Forests were identified as two or more enhancer trees connected via shared nodes (Table S7). To annotate these trees, we integrated the ATAC tag density to the nodes, and the HiChIP PETs to the edges of the cell-type-specific enhancer trees. Cytoscape[92] was used to visualize the annotated trees for select genes in relevant cell types, with node size representing ATAC tag density and edge thickness reflecting HiChIP PET counts.

### Analysis of enhancer-promoter distances and skipped nearest promoters

To quantify the linear genomic distance between enhancers and their associated promoters, we identified all enhancer chains following a $P_0$-$E_1$-$E_2$ pattern, where a promoter (P) connects to a first enhancer ($E_1$), and $E_1$ in turn connects to a second enhancer ($E_2$). For each chain, we calculated the genomic distances from the midpoint of $E_1$ nodes to the TSS ($d_{E_1}$) or from the midpoint of $E_2$ node to the TSS ($d_{E_2}$). We also quantified HiChIP interaction strength as paired-end tag (PET) counts for $L_1$ and $L_3$ edges.

To determine whether enhancers preferentially loop to their nearest promoter, we compared the linear distance between each enhancer (E1) and its looped promoter (Lp.P) against the distance between that enhancer and its nearest promoter (Nrst.P). The nearest promoter was defined as the closest TSS located within 5 kb to 1 Mb from the enhancer midpoint, irrespective of looping. The *annotatePeaks.pl* program from the Homer suite version 5.1 was used to calculate the distances between neighboring genes and tree nodes. Enhancers for which looped promoter differed from their nearest promoter were classified as "skipping" their nearest promoter. For these cases, we annotated the skipped and looped promoters with gene type as coding vs. noncoding using *annotatePeaks.pl* program from the Homer suite version 5.1. We further compared expression status for each pair of looped and skipped promoters for both cell type specificity and abundance of their transcripts, using published pancreas single cell RNA-seq datasets.[50] Specifically, we identified which gene (looped vs. skipped) had higher cell-type specificity and transcript abundance (Table S6). We further classified skipped promoters according to their genomic position relative to the enhancer loop as either InsideLoop—skipped gene and looped gene are both located either upstream or both downstream of the enhancer node, or OutsideLoop—skipped gene and looping gene are located on opposite sides of the enhancer node, meaning one gene is upstream and the other downstream relative to the enhancer node. These classifications were illustrated in Figure 2D. The counts of skipped promoters in each category were tabulated for each cell type (Table S6). To test whether the distribution of InsideLoop vs. OutsideLoop was uniform across cell types, we performed a chi-square test ($\chi^2 = 7.21$, df = 4, $p = 0.125$), indicating no significant difference in orientation bias.

### Evaluation of enhancer importance by *in silico* perturbation (EPIC)

To estimate the effect size of individual enhancers in an enhancer-promoter tree, we developed a machine learning method to score enhancers by iteratively removing one enhancer at a time and tracking the change in the accuracy of the trained model.

### Preparing the input data

We selected four sets of enhancer-promoter trees corresponding to four cell types (α-cell, β-cell, acinar, and duct). Data for δ-cell were excluded due to substantially fewer number of enhancer trees detected (Figure S2A). δ-cells are among the least abundant cell types within islets (less than 100,000 cells per donor). While we were able to detect frequently occurring loops, like those in the *SST* locus, the low cell number precludes constructing HiChIP libraries with complexity comparable to the other cell types. We used the expression specificity scores (ESS)[50] to subset enhancer-promoter trees that represent both cell-type-specific and not cell-type-specific genes. Based on our prior work, ESS greater than 0.7 indicates high cell-type specificity, whereas ESS less than 0.3 indicates low specificity.[50] In the α-cell dataset, there were 354 trees whose associated genes had an ESS greater than 0.7, therefore these trees were assigned the class value "alpha". To ensure a balanced representation of the non-specific class, we randomly selected a similar number of genes whose ESS is less than 0.3 in all four cell types and assigned them as "non-alpha". We applied a similar approach for the other cell types. For β-cell versus non-β-cell, we used 340 trees for "beta" and 365 for "non-beta". The acinar versus non-acinar comparison included 641 trees for "acinar" and 652 for "non-acinar". Finally, for duct versus non-duct, we used 476 trees for "duct" and 447 for "non-duct". We named these sets as $S_{alpha}$, $S_{beta}$, $S_{acinar}$, and $S_{duct}$. We then built a matrix containing predictor variables based on our chromatin interaction and accessibility data corresponding to these selected enhancer trees. Specifically, for a given tree that belongs to the cell-type A, the sum of ATAC-seq tag densities of direct enhancer nodes (ATAC-*d*-A), the sum of HiChIP PET counts of direct edges (PET-*d*-A), the sum of their products (ATACxPET-*d*-A), and similar 3 variables for indirect nodes (ATAC-i-A, PET-i-A, and ATACxPET-i-A). In total, 24 variables of chromatin derived data were included in the matrix.

### Initial modeling and performance evaluation

We employed the k-Nearest Neighbor (kNN) algorithm to classify the cell types that the enhancer tree promoters belong based on the predictor variables detailed above. The data were centered and scaled to normalize, and Euclidean distance was used to measure the distance between data points. 10-fold cross-validation was employed to identify the optimal number of neighbors (k). We built the original models $M_{alpha}$, $M_{beta}$, $M_{acinar}$, and $M_{duct}$ on the $S_{alpha}$, $S_{beta}$, $S_{acinar}$, and $S_{duct}$ sets. To evaluate the model performance, we used the accuracy metric, which is the ratio of correct observations (true positives) to the number of total observations.

### Comparison of the original tree model performances to alternative genomic models

We compared the performance of tree models against two other models: Promoter accessibility model and the linear model. Promoter accessibility model was built by taking only ATAC-seq tag counts at the promoters and no further chromatin information such as PET counts or interacting enhancers was considered. We defined the promoter regions as 2 kb upstream and 3 kb downstream of the transcription start site (TSS). In the linear model, we assigned a 'gene regulatory domain' that extends 1 Mb both upstream and downstream of the TSS. For this model, only the ATAC-seq tag counts of enhancers within this domain were considered, and the ATAC-seq density of each neighboring enhancer within such window was scaled to be inversely proportional to its genomic distance to the promoter.

### Enhancer removal and evaluating the perturbed model performances

For each enhancer node 'e' of the tree 'T', we created perturbed models $M_{alpha}\setminus\{e\}$, $M_{beta}\setminus\{e\}$, $M_{acinar}\setminus\{e\}$, and $M_{duct}\setminus\{e\}$ by removing the enhancer node 'e' and its child nodes. All 24 predictor variables (ATAC-seq tags, HiChIP PET counts, their products [ATACxPET] for four cell types, direct/indirect node) were recalculated after each enhancer removal. Similar to the original models, we used the kNN algorithm with 10-fold cross-validation splits. By comparing the accuracy difference between the original model and the perturbed one, we calculated the deviation of prediction accuracies between e-deleted tree and the original prediction for each enhancer tree set. By summing up the absolute values of these accuracy deviations across cell types, the total deviation of the enhancer e is given as

$$D(e) = \sum_{M \in \{M_{alpha}, M_{beta}, M_{acinar}, M_{duct}\}} |acc(M) - acc(M\setminus\{e\})|$$

where acc stands for the accuracy of the learning model. We repeated these procedures 100 times for each enhancer to obtain a robust estimate of the average total deviation. This average was defined as the 'effect size' for tree T. Enhancers with larger total deviations were considered to have a greater impact on their associated promoter activity. We implemented EPIC algorithm in R, using the caret library for model performance evaluation.

## Perturbing enhancers using CRISPR interference in human pancreas cells

### CRISPR guide RNA design

CRISPR guides were designed using the software Geneious Prime 2023.0.1 (https://www.geneious.com) and were selected based on activity[95,96] and specificity scores that exceeded 94%.[97] We targeted each anchor region (enhancer node) with minimum three gRNAs. For the complete list of guides see Table S11.

### Production of adenoviral vectors and viral stocks

Adenoviruses carrying the CRISPR constructs were provided by Vector Biolabs using their custom adenovirus construction service (https://www.vectorbiolabs.com/). Ad-CMV-NLS-dCas9-VP64-mCherry (CRISPRa) was purchased from Vector Biolabs. For CRISPRi, Ad-CMV-NLS-dCas9-KRAB-mCherry were cloned by modifying the SFFV-dCas9-KRAB-BFP (addgene46911) at Vector Biolabs then packaged into adenoviruses. For gRNA viruses, gRNA arrays for each target region were assembled into CARGO constructs following the previously described protocol,[84] these constructs then were sent to Vector Biolabs to produce adenoviruses. All virus particles were produced at a minimum titer of $1 \times 10^{10}$ PFU/ml (PFU, plaque-forming units).

### Adenoviral transduction of primary human pancreas cells

Human islets or exocrine tissues were partially dissociated using enzymatic digestion as described previously (Accutase, Thermo Fisher Scientific).[39] Dispersed cells were seeded on AggreWell400 wells (STEMCELL 34415) at an estimated 300,000–500,000 cells per microwell containing the appropriate culture media. The islet culture media consisted of RPMI 1640 with Glutamine, supplemented with 10% fetal bovine serum (FBS) (HyClone) and 1% penicillin/streptomycin (P/S) (Thermo Fisher Scientific 15140122). The exocrine culture media was composed of CMRL 1066 supplemented with 10% heat-inactivated FBS, 1% GlutaMAX (Thermo Fisher Scientific 35050061), 1% P/S, and 0.1% nicotinamide (Sigma-Aldrich 72340). Cells were transduced with adenovirus at an MOI of 100 and cultured at 37°C for 2 h. After the incubation, viral media were removed, cells were washed with PBS twice and returned to the incubator with fresh media to allow aggregation. Post-transduction, the cell clusters were transferred to a 24-well ultra-low attachment plate (Corning 3473) on day 3. The media was refreshed as needed, and the cells were harvested for hybridization chain reaction (HCR) preprocessing on day 5.

## RNA-FISH using hybridization chain reaction (HCR) in primary human pancreas cells

CRISPR treated and control cell clusters were collected and dissociated into monolayer using enzymatic dispersion (Accutase, Sigma-Aldrich A6964). After fixation with 4% paraformaldehyde at room temperature for 1 h, the cells were washed and stored in 1% BSA in PBS with Riboblock (Thermo Fisher Scientific EO0384) at 4°C until further processing. For permeabilization, cells were

treated with PBS-Tween20 (PBS-T) containing Riboblock, followed by storage in 70% ethanol at −20°C. The hybridization was performed with probes specific to the targeted genes, followed by a series of washes to remove unbound probes. Amplification of the hybridized probes was carried out using hairpin oligonucleotides, and the samples were stained with DAPI for nuclear visualization. Hybridization experiments were performed using HCR RNA-FISH bundles from Molecular Instruments, including customized probe sets, amplifiers, and buffers specific for single cell suspension and staining. For the list of probes, see Table S12. For each sample, multiplexed staining was performed using combinations of target gene probes and marker gene probes. Matched fluorescently labeled amplifier hairpins were enabled specific quantification of gene transcripts within the appropriate cell types. The cells were counterstained with DAPI (4′,6-diamidino-2-phenylindole) and seeded at 70–90% density (50,000–75,000 cells per well) on collagen-coated, optically clear 96-well imaging plates (Revvity Health Sciences, D6055700) immediately prior to imaging.

### High-throughput imaging and quantitative analysis

HCR samples were imaged in four channels (405, 488, 561, and 640 nm) using a dual spinning disk high-throughput confocal microscope (Yokogawa CV7000 or CV8000). A 60× water immersion objective (NA = 1.2) was used for imaging monolayer samples. Two 16-bit sCMOS cameras were employed with binning set to 1, yielding a pixel size of 108 nm. Image Z-stacks were acquired at 0.5-micron intervals across a total depth of 8 microns. For each well, 20–48 randomly selected fields (containing approximately 10–100 cells per field) were imaged. Images were automatically corrected in real-time using Yokogawa's proprietary software to address camera alignment, optical aberrations, vignetting, and camera background issues. The maximally projected and corrected images were saved as 16-bit TIFF files. Quantitative analysis of the HCR results was done using Columbus 2.9.1 (PerkinElmer/Revvity) or Signal Imaging Artist (SiMA, PerkinElmer/Revvity) 1.2 software. The primary output was the mean fluorescence intensity of RNA-FISH puncta in the far-red channel (640 nm, Alexa 647) measured over individual cell bodies. This channel corresponded to gene targets including *PCSK1*, *PCSK2*, and *XBP1*. Cell-type-specific markers (*INS*, *GCG*, *CPA1*, *SPP1*) were labeled with Alexa 488 and detected in the 488 nm channel. Additional probe and barcode information is provided in Table S12. Single-cell results were exported from Columbus or SiMA as tabular text files. These results are provided for all donors in Table S9. Statistical analysis and data plotting were done in GraphPad Prism version 10.2.3 (GraphPad Software, USA, www.graphpad.com).

### GWAS risk SNP enrichment analysis in cell-type-specific enhancer trees

### GWAS data selection

GWAS summary statistics for traits associated with pancreas disorders and control traits were obtained from various key studies: type 2 diabetes,[66] fasting glucose, fasting insulin, glycated hemoglobin and 2-h glucose[65] type 1 diabetes,[32] and pancreatic ductal-adenocarcinoma (PDAC),[64] control traits.[98,99] GWAS summary statistics were downloaded from Type 2 diabetes Knowledge portal (https://t2d.hugeamp.org/), MAGIC consortium (http://magicinvestigators.org/downloads/), GWAS-EBI catalog (https://www.ebi.ac.uk/gwas/). GWAS summary statistics for PDAC were provided by the Pancreatic Cancer Cohort Consortium and the Pancreatic Cancer Case-Control Consortium.

### GARFIELD analysis

We used GARFIELD algorithm[67] to calculate the enrichment of GWAS variants that overlapped with our cell type specific enhancer trees. Specifically, we used cell-type-specific ATAC-seq peak coordinates (Table S3) that overlap with enhancer tree nodes (enhancers and promoters), and prepared annotation overlap files as required by the algorithm at different GWAS thresholds.

We used the data on LD tags, distance to TSS, and minor allele frequency provided by GARFIELD, which implements LD scores based on the UK10K project as the reference set for European population. GARFIELD enrichment tests were run individually using summary statistics from each of the GWAS studies listed above, and the coordinates for all four cell types (α, β, acinar, and duct) defined above were used as input annotations. The estimated odds ratios (ORs) and enrichment $p$ values were computed at various GWAS $p$ value thresholds ($1 \times 10^{-4}$, $1 \times 10^{-5}$, $1 \times 10^{-6}$, $1 \times 10^{-7}$, $5 \times 10^{-8}$, and $1 \times 10^{-8}$). The R code `Garfield-Meff-Padj.R` from GARFIELD was used to calculate an enrichment $p$ value threshold. This threshold was adjusted using Bonferroni correction ($p < 0.01$) to account for multiple testing, based on the effective number of annotations (Meff = 4.45). The negative log2 of enrichment $p$ values for the tested GWAS summary statistics were plotted using ggplot package in R.

### SNP-to-target identification and evaluating EPIC effect sizes

We used significantly enriched SNPs from the $p < 1 \times 10^{-4}$ GWAS threshold which was the largest bin containing the highest number of enriched SNPs from all four GWAS $p$ value thresholds. While this is a relatively lenient GWAS $p$ value threshold, we wanted to include the maximum number of enriched SNPs in the beginning for our gene mapping analysis. We then identified the overlapping SNPs by intersecting the SNP coordinates with enhancer tree node coordinates using BEDtools. Each overlap was represented with a unique identifier for each SNP-enhancer node-promoter link (Table S10). The list was further filtered to include enhancer trees containing at least one enriched SNP node, and in some instances, has cell-type-specific promoters with an ESS > 0.7.[50] We examined three pancreatic traits (type 1 diabetes, type 2 diabetes, and pancreas cancer) across three cell types (acinar, duct, beta), comparing over 7000 enhancers and 300 genes (see table below).

| | PDAC duct | PDAC acinar | T2D acinar | T2D beta | T1D acinar | T1D duct | total |
|---|---|---|---|---|---|---|---|
| Number of enhancers | 621 | 359 | 3356 | 1971 | 649 | 564 | 7520 |
| Number of genes | 31 | 9 | 172 | 30 | 45 | 41 | 328 |

We used EPIC to determine the effect size of all enhancer nodes that were included in these lists. Next, the EPIC evaluated nodes were stratified into two groups: SNP-overlapping or not-overlapping, and the effect sizes of these nodes were represented by group using CDF() function in R.

### Visualization of select SNP-target gene regions

Examples of SNP-target gene networks for T2D and PDAC in acinar cells were visualized using locuszoom (https://my.locuszoom.org/) and the UCSC genome browser (https://genome.ucsc.edu/) on hg38 genome build.

