## [Document S2. Transparent peer review records for Wang et al. · Cell Genomics]

Predictive Prioritization of Enhancers Associated with Pancreas Disease Risk

Li Wang, Songjoon Baek, Gauri Prasad, John Wildenthal, Konnie Guo, David Sturgill, Thuchhi Truongvo, Erin Char, Gianluca Pegoraro, Katherine McKinnon, The Pancreatic Cancer Cohort Consortium, The Pancreatic Cancer Case-Control Consortium, Jason W. Hoskins, Laufey T. Amundadottir, H. Efsun Arda

Summary

Initial submission: Received : Nov 01, 2024

Scientific editor: Sara Rohban

First round of review: Number of reviewers: 3
Revision invited : Jan 16, 2025
Revision received : Jul 14, 2025

Second round of review: Number of reviewers: 3
Accepted : Sep 19, 2025

Data freely available: YES

Code freely available: YES

This transparent peer review record is not systematically proofread, type-set, or edited. Special characters, formatting, and equations may fail to render properly. Standard procedural text within the editor's letters has been deleted for the sake of brevity, but all official correspondence specific to the manuscript has been preserved.

Referees' reports, first round of review

Reviewer #1:

The manuscript presents a comprehensive study of enhancer-promoter interactions in human pancreas cells using 3D chromatin conformation assays and a novel tree-based modeling approach. The researchers developed an algorithm to predict the impact of enhancer perturbations on gene expression, linking key enhancers and noncoding genetic variants to cell-type-specific transcription and pancreas disease risk. Validation experiments using CRISPR interference highlighted the critical role of acinar cell enhancers in pancreatic ductal adenocarcinoma are also included.

Overall, this manuscript presents an innovative approach to studying tissue-specific enhancers and their role in disease risk. The manuscript is well-written, but some sections are overly technical and could be refined to convey the message more clearly while simplifying the technical details.

The figures could be streamlined and reduced in number and panels.

For example in Figure 1 the examples are interesting while the PCAs (B and C) are important but could be moved to supplementary material

In F, using a log-transformed Y-axis could enhance the visualization of changes in the regions where the most abundant data is concentrated.

Figure 3 A, B and C are mainly descriptive and I would suggest to move them to the supplementary material

Figure 4 A is a scheme that is informative but could be moved to supplementary

A direct comparison of the enhancer forests with enhancer hubs (10.1038/s41588-019-0457-0), despite the latter not being defined in a cell type-specific manner, could reveal whether there is a correspondence between the two concepts and aid in interpreting the results.

The last paragraph: "Enhancer trees and EPIC facilitate functional annotation of genetic variants associated with pancreas diseases" is possibly the most important part of the story but some additional statistical analysis could strengthen the findings:

Are enhancers predicted by EPIC to have higher effect sizes more enriched with trait-specific GWAS SNPs compared to those with lower EPIC scores? The effect size distributions of enhancers overlapping and non-overlapping SNPs could be for example plotted and statistically compared. A few additional example to span the different cell types including a functional validation could add strength to the overall message.

Reviewer #2:

In this manuscript, Wang and colleagues report exciting efforts that combine innovative epigenomic profiling, analytical tools and approaches, and CRISPR/Cas9-based epigenome editing tools to define and interrogate pancreas cell type-specific cis-regulatory network connectivity in health and disease. Using tree-based network models, they establish cis-regulatory element modules that reveal important, and somewhat surprising, regulatory principles of transcriptional regulation in endocrine (alpha, beta, and delta) and exocrine (ductal, acinar) pancreas cell types. The structure and connectivity of these tree-based modules revealed a surprising propensity for enhancers in each cell type to skip the nearest gene promoter and instead loop to one more distal (>3/4 of detected loops!!) and a largely promoter-centric connection between distinct enhancer modules. The authors leverage these tree-based networks and

published single cell transcriptome profiles of the corresponding cell types to build models predicting the relative importance of each enhancer in their respective regulatory modules to cell type-specific target gene expression. They complete exciting CRISPR-based epigenome perturbation experiments in primary human islet and acinar cell types to test and validate multiple predictions. By assessing disease-associated sequence variant enrichment in and potential molecular effects on these enhancers, they nominate putative causal/functional variant-target gene relationships and uncover a general principle in which these SNPs overlap enhancers with larger predicted effects than those not containing SNPs. Moreover, they highlight potential roles for unanticipated cell types in disease susceptibility. For example, they a pancreatic ductal adenocarcinoma-associated SNP overlapping an acinar cell type enhancer and demonstrate that its CRISPR epigenetic inactivation alters XBP1 expression, implicating altered acinar cell ER stress as a potential molecular and cellular mechanism underlying genetic risk for PDAC. This study is very exciting, and the data and insights presented advance our understanding of pancreas cell type-specific transcriptional regulatory networks and key pieces of information, such as potential functional variants and their target genes, important for variant-to-function translation of GWAS association signals into biological mechanisms. Pending the authors' ability to address some minor to moderate questions and concerns that arose during review, I believe this will be an exciting and impactful paper that will appeal to the readership of Cell Genomics. Questions and suggestions, divided into major and minor categories, are as follow:

Major questions/suggestions:

- 1) The HiChIP data as presented in the initial figures looks INCREDIBLY clean, suggesting that there are no loops at highlighted marker gene regions for each cell type in Figure 1C-H and related supplementary. Is this true that there are really no loops detected in non-target cell types? Typically, there is at least some level of looping, not necessarily specific or similar to that in the target cell type, observed in the others. Similarly, it is important to see the ATAC-seq profiles and peak calls in the other cell types. As it stands, it is impossible to objectively evaluate the quality and reproducibility of these exciting and critical profiles generated.
- 2) Related to question 1 above, how reproducible or robust are the ATAC-seq peaks and Hi-CHIP loops between the donor replicates for each cell type studied? It is important to provide some more targeted assessment and visualization of peak and loop calls between each biological replicate of the enriched cell types profiled. The Principal Component plots suggest clear overall distinctions between the cell types, but additional measures of correlations between replicates, such as pairwise correlation plots comparing within-cell type, between-donor variation vs. between cell type variation or irreproducible discovery rate (IDR) analysis, is needed.
- 3) Delta cell peaks and loops are strikingly lower than other cell types despite the apparent purity and profiles shown in Figures 1 and S1. Is this related to the ultimate yield of the sorted samples being low? Could the authors provide more details on relative yield, in addition to the compelling and impressive 95% cell purity, that they achieved? This would seem to warrant more discussion/comment in the text surrounding Figures 1, S1, or S2A. It also raises some questions about the very small delta cell enhancer trees and lack of expression specificity and transcript abundance enrichment trends observed in other cell types and highlighted in Figures 2F and Supplementary Figure 2I
- 4) The observation that most enhancers skip their nearest linear distance gene promoter is very surprising to this reviewer. Could some of this apparent skipping be due to the certain limits in resolution to detect looping interactions between nearby (<5 kb, <10 kb) cis-regulatory elements? In essence, is there a bias of short or shorter distances for the "apparently skipped" nearest promoters? Or orientation biases driving these relative percentages? I.e., what is the relative proportion of skipped nearest promoters on the opposite side of the potential loop (left-hand example in Figure 2D "loop-to-

distal" cartoon) vs. those in which the skipped nearest promoter is within the loop?

5) The tree/forest quartiles are really interesting in terms of their relative enrichment for expression specificity and increased transcript abundance. Are there unique or interesting transcription factor binding motifs (TFBMs) or other sequence features that distinguish or are enriched in the large vs. small tree enhancer "forests"? In essence, are there distinct transcription factors or general DNA binding factors that nucleate or stabilize large vs. small enhancer modules identified?

6) Approximate correlations in enhancer forest and TAD genomic span sizes is intriguing. Can the authors take this a step further and compare specific enhancer forest spans to their corresponding TADs in the genome? I.e. do the enhancer forests overlap TADs in each of the chromosome regions? Or is there no clear correlation between enhancer forest spans and TADs per location?

7) Are the apparent cell type-specific tree structures and connections unique to the cell type vs. its related endocrine or exocrine counterparts? I.e., alpha-specific vs. all endocrine or ductal cell vs. all exocrine? Or is this a one-vs-all comparison? Are these apparent specificity features driven by the strong out-group comparisons (for example beta vs. acinar or ductal) rather than the similar cell types (beta vs. alpha, delta)? Are there interesting themes and insights that emerge from 'endocrine' vs. 'exocrine' comparisons? Or 'within-compartment' comparisons?

8) Related to 7 above, this reviewer is very curious if these distinctive enhancer modules or forests reveal differential gene set or pathway enrichments? What about motifs? Do these distinctive modules reveal key (and different) transcription factors that might nucleate or stabilize these distinct enhancer forests?

9) It was surprising and not particularly intuitive to read that the linear and full tree alpha cell models performed similarly and suggested a distinct organization of alpha-cell specific genes in the genome. Can the authors expand upon this observation and assertion and comment or speculate on why this might be the case and what this might mean? Is this apparent distinct alpha-cell specific gene organization reflected in other data collected, such as the number of peaks or the proximity or density of alpha-cell interactions vs. the other cell types?

10) CRISPR enhancer targeting in Figure 5 is innovative and exciting! Can the authors also show in the main or supplementary figures the per-donor targeting effects? Since human islets can be variable in terms of behavior, quality, etc., it will take the robustness and reproducibility of these observations even further by seeing it is consistent between the two or more independent donors tested according to the figure legend. Additionally, it would be informative to include representative images of the promoter-targeting gRNA experiments alongside the selected enhancers in Figures 5E and 5H.

11) Where do the enhancers to the right of PCSK2 in Figure S4D rank? I.e., are those predicted to be stronger enhancers but just not specific? Or are they poor-ranking candidates?

12) It was surprising that enhancer targeting with the strong VP64 activator was more powerful than promoter targeting. Why do the authors think this is the case? Is it possible these strong enhancers may serve as other promoters? Or are they very clear intragenic or intergenic enhancers? Are there other features of these enhancers that might predict their stronger regulatory effects? Sequence motifs? High eRNA levels vs. low?

13) Related to 12 above, why do the authors think similar targeting didn't activate expression in exocrine cell types? Do chromatin states or ATAC-seq maps give a clue? Why does targeting the gene promoter with strong VP64 transactivator not even elevate gene expression in these cell types?

14) The observation that SNP-containing enhancers have higher predicted EPIC effect sizes was VERY interesting. While I appreciate the focus on unexpected phenotypes and cell types for the GWAS enrichment studies, it seems really important to include the type 1 and type 2 diabetes analyses and interpretations before switching to acinar and cancer connections. Given the extensive focus on endocrine cell type features and analyses, this switch seemed abrupt and not well-justified. I strongly encourage further inclusion and work-up of diabetes and related phenotype results with the endocrine

cell type maps and models in the main and supplementary figures. This seems to be an important missed opportunity to expand our knowledge and understanding of diabetes GWAS target genes and potential regulatory features and insights.

15) In Supplementary Figure 5A-C, it would be helpful to show the corresponding data for the other cell types so one can compare their enhancer predictions and maps with their respective S5C expression.

16) The switch from CRISPRa assessment for Figure 5 PCSK enhancer experiments to CRISPRi for Figure 6 acinar enhancer experiments are not well justified. Was there a particular reason to apply consistent perturbations between these experiments? Or to show for one or both complementary effects of CRISPR activation vs. inhibition? Was CRISPRi selected because the PDAC risk allele is predicted to decrease enhancer accessibility, activity, or target gene expression?

17) The authors' implication of XBP1 as an acinar cell-specific expressed gene is odd given well-documented roles in beta-cell and islet protein folding and endoplasmic reticulum stress. Also, it is unclear how or why Figure 5E suggests it has a higher expression specificity score in acinar cells although it appears expressed (at comparable levels) in alpha and beta cells. Is the acinar cell specificity of this variant and XBP1 regulation better supported by comparing and contrasting the regulatory profiles and enhancer forests in acinar vs. the other cell types profiled? Can this be shown in main or supplemental figures to better document and motivate the specific selection of acinar cells as the cell type of interest and importance? This should also apply to Supplementary Figure 5 so one can assess the relative structures and specificity of GATA4 trees, profiles, and loops.

18) In the Discussion second-to-last paragraph, perhaps the authors could include consideration and discussion of the potential for monogenic disease mutations or other rare variants to be present and identified.

Minor questions/revisions:

1) Text is confusing as to how many donors/samples have been profiled. In the abstract it indicates 20 but end of Intro paragraph indicates 27. It would be helpful if the authors could be more transparent, clear, and consistent as to how many individuals' profiles the data represent.

2) Figure 2E and other relevant expression specificity score legends: define "expression specificity score" prior to the abbreviation ESS. It appears the "score" or other term for the second "S" in ESS is missing from the definitions in the legends.

3) Supplementary Figure 2G: "loop-skp gene pairs" should be changed to "loop-skip gene pairs"

4) Page 7, change "Supplemental Figure 2B-D" to "Supplementary..." to be consistent with other references to supplementary figures. This appears to be an occasional recurrent classification difference that should be checked and harmonized throughout the text.

5) Figure 5B and related transduction efficiency evaluations: can the authors provide FACS or other more quantitative measures and metrics? It is entirely possible that alpha and beta cells could exhibit differences in transduction efficiency.

6) Figures 5F, 5I: y-axis values appear to be missing for alpha cells. Is it the same as betas? If so, the authors should plot them all together on one large scatter plot. If not, it is important to include the different y-axis values so one can compare the relative magnitude of effects these epigenetic perturbations elicit.

7) Discussion, page 18:

a. "that most enhancers" phrase is duplicated. One instance should be deleted

b. Change "potentially an additive model" to "potentially additive, model"

c. Suggest to change "potentially, already positioned for transcription" to "potentially pre-positioned for transcription..."

d. Change "like HiChIP is they capture chromatin" to "like HiChIP is that they capture..."

e. Remove comma following references 73,74

Reviewer #3:

Wang et al reported the development of a new strategy to prioritize enhancers in cells related to islets and their association with pancreas disease risk. Authors performed H3K27ac HiChIP and ATAC-seq experiments in cells sorted from pancreas organ donors. They then build a graph-based network for enhancer-enhancer interactions. They studied the properties of the tree across different cell types. They then used machine learning to predict how perturbation in the enhancer network controls gene expression. Finally, they used RNA FISH and validated predictions of their machine learning using CRISPRa and CRISPRi, adding or removing enhancer activities.

This study is very rigorous, cell types are purified from organ donors and experimental set up is well suited for studying enhancer-enhancer network analysis. The computational approach to decipher the complexity of enhancer interaction is innovative and well justified. Validation data using RNA FISH increases the confidence for machine learning predictions. I have two questions for the authors:

- 1) There is an opportunity to define the nature of genes with distinct enhancer topology across different cell types. For example what are the specific genes with high level of enhancer connectivity? do transcription factors, genes related to metabolism or other categories have high connectivity? What is the nature of genes with low connectivity? Are these patterns different across cell types?
- 2) I recommend authors to create a web portal and enable the community to browse through the enhancer network connectivity across different cell types. This type of web portal can have a broad impact enabling pancreas researchers to test different hypotheses using this valuable resource.

Authors' response to the first round of review

We are grateful to all reviewers for their thoughtful and constructive feedback. The suggestions provided have significantly strengthened the manuscript, and we deeply appreciate the time and care taken to evaluate our work. It was especially rewarding to read the enthusiasm for the dataset and the analysis framework, and we hope that the experimental and computational resources we have generated will be valuable to the broader scientific community. We thank the reviewers and editors again for the opportunity to revise and improve this study. Below are our responses to the specific comments:

Reviewer #1:

The manuscript presents a comprehensive study of enhancer-promoter interactions in human pancreas cells using 3D chromatin conformation assays and a novel tree-based modeling approach. The researchers developed an algorithm to predict the impact of enhancer perturbations on gene expression, linking key enhancers and noncoding genetic variants to celltype-specific transcription and pancreas disease risk. Validation experiments using CRISPR interference highlighted the critical role of acinar cell enhancers in pancreatic ductal adenocarcinoma are also included.

Overall, this manuscript presents an innovative approach to studying tissue-specific enhancers and their role in disease risk. The manuscript is well-written, but some sections are overly technical and could be refined to convey the message more clearly while simplifying the technical details.

- 1- The figures could be streamlined and reduced in number and panels. For example in Figure 1, the examples are interesting while the PCAs (B and C) are important but could be moved to supplementary material

We thank the reviewer for the suggestion. We moved the PCA plots to the supplement, now in **revised Figure S1D, F**.

- 2- In F, using a log-transformed Y-axis could enhance the visualization of changes in the regions where the most abundant data is concentrated.

We appreciate the feedback. We considered applying a log-transformation to the ATACseq data to better visualize regions with high peak density. However, log-transforming the data substantially compressed differences in peak height, making visually discernable differences appear smaller. For instance, a d-cell peak with 45,000 tags and a b-cell peak with 500 tags (a 90-fold difference) would appear as 15.46 vs. 9 after logtransformation, visually showing less than a ~2-fold difference. Since Reviewer #2 also had feedback on Figure 1, we generated an alternative visualization by broadening the genomic regions to include non-cell type-specific genes (**revised Figure 1**). We think this new figure highlights differences in peak height, chromatin accessibility, and HiChIP loops across cell types at specific and non-cell type-specific loci, addressing the reviewer's comment.

- 3- Figure 3 A, B and C are mainly descriptive and I would suggest to move them to the supplementary material

We appreciate the reviewer's suggestion and carefully revisited Figure 3. We agree that panel A, which summarizes general forest features, is primarily descriptive and have moved it to the supplement (now **revised Figure S3D**). However, we think that panels B and C demonstrate key features of the enhancer tree models—namely, the prevalence of promoter-promoter interactions and enhancer-enhancer connectivity patterns that are not captured by standard loop-by-loop pairwise analyses. These panels support the rationale for the subsequent modeling and prioritization work. We have updated the figures and legends accordingly in the revised manuscript.

- 4- Figure 4 A is a scheme that is informative but could be moved to supplementary

We thank the reviewer for suggesting ways to streamline technical content. We think Figure 4A helps readers understand the EPIC model and its underlying data structure by visually summarizing how enhancer-promoter trees are categorized into cell typespecific and non-specific groups, and how chromatin accessibility, interaction strength, and enhancer activity inform the modeling framework. For these reasons, we have retained Figure 4A as a main figure. However, we agree that the model comparisons in Figure 4B-C were overly technical and peripheral to EPIC's enhancer prioritization, and have therefore subordinated these panels to the supplement (**revised Figure 4 and S4**).

- 5- A direct comparison of the enhancer forests with enhancer hubs (10.1038/s41588-0190457-0), despite the latter not being defined in a cell type-specific manner, could reveal whether there is a correspondence between the two concepts and aid in interpreting the results.

We thank the reviewer for suggesting a comparison of our enhancer forests with the “enhancer hubs” described by Miguel-Escalada et al. (PMID: 31253982). In their paper, the authors used promoter capture Hi-C on whole islets to define promoter-associated 3D spaces. However, their analysis was explicitly constrained within TAD-like boundaries derived from their data, potentially limiting the resolution and range of TADs compared to standard Hi-C methods. Nonetheless, per reviewer’s request, we performed a direct overlap analysis between the enhancer hubs defined in MiguelEscalada et al. (Supplementary Data Set 5) and our enhancer forests. We found that nearly 100% of their enhancer hubs lie within our enhancer forests. To quantify the degree of overlap, we calculated a Jaccard Index (overlap region divided by union of forest and hub regions). We observed a median Jaccard index of ~0.15, meaning that our forests span much larger genomic intervals and contain more promoters (median 7 vs. their median 2). We attribute these differences to using HiChIP vs promoter-capture Hi-C and our network approach of merging trees based on connectivity without imposing additional structural restrictions such as TAD boundaries or a minimum enhancer count. Even though the specific descriptions are different, we think the high-level conclusion is consistent with both datasets—genes central to islet (pancreas) function and cell typespecific expression exhibit more 3D enhancer connectivity. We have added a brief note clarifying these distinctions in the text:

“A similar principle was observed in a study (Miguel-Escalada et al. 2019) using human islets and promoter capture Hi-C to show dense 3D enhancer connectivity at genes critical for islet cell function.”

- 6- The last paragraph: "Enhancer trees and EPIC facilitate functional annotation of genetic variants associated with pancreas diseases" is possibly the most important part of the story but some additional statistical analysis could strengthen the findings: Are enhancers predicted by EPIC to have higher effect sizes more enriched with trait-specific GWAS SNPs compared to those with lower EPIC scores? The effect size distributions of enhancers overlapping and non-overlapping SNPs could be

for example plotted and statistically compared. A few additional examples to span the different cell types including a functional validation could add strength to the overall message.

We appreciate that the reviewer finds our work valuable and thank the reviewer for highlighting the key aspect of our study. As requested, we compared EPIC effect sizes for enhancers overlapping trait-associated GWAS SNPs and with those without overlap. We visualized these comparisons as cumulative distribution function (CDF) plots and box plots (**revised Figure 6B and S7**). We now report P -values from Kolmogorov–Smirnov (KS) tests, with all comparisons showing significant divergence between the distributions (P -value < 0.05). Enhancers overlapping trait-associated variants consistently show higher EPIC effect sizes. These findings are now explicitly noted in the revised Results section (page 15, lines 350-359) and expanded in Methods. For this revised analysis, we examined three pancreatic traits (type 1 diabetes, type 2 diabetes, and pancreas cancer) across three cell types (acinar, duct, beta), comparing over 7000 enhancers and 300 genes, summarized in the table below. We think these sample sizes provide reasonable power to support our statement that enhancers enriched with traitspecific variants tend to have higher EPIC effect sizes.

	PDAC duct	PDAC acinar	T2D acinar	T2D beta	T1D acinar	T1D duct	total
Number of enhancers	621	359	3356	1971	649	564	7520
Number of genes	31	9	172	30	45	41	328

Regarding functional validation, our study includes perturbation experiments on three target genes, spanning 12 enhancer regions, 36 different guide RNAs, tested on different cell types and in tissues from at least two donors. To our knowledge, this represents one of the most comprehensive enhancer validation efforts performed in primary human pancreas cells. These experiments demonstrate that transcriptional responses to enhancer perturbation scale with EPIC-predicted effect size. While we agree that more examples would always add strength, these assays are highly resourceintensive (donor tissue availability, virus generation, data acquisition and analysis), and additional experiments are not feasible within a reasonable revision timeline.

We thank the reviewer for prompting us to clarify and strengthen our presentation of these key findings.

Reviewer #2:

In this manuscript, Wang and colleagues report exciting efforts that combine innovative epigenomic profiling, analytical tools and approaches, and CRISPR/Cas9-based epigenome editing tools to define and interrogate pancreas cell type-specific cis-regulatory network connectivity in health and disease. Using tree-based network models, they establish cisregulatory element modules that reveal important, and somewhat surprising, regulatory principles of transcriptional regulation in endocrine (alpha, beta, and delta) and exocrine (ductal, acinar) pancreas cell types. The structure and connectivity of these tree-based modules revealed a surprising propensity for enhancers in each cell type to skip the nearest gene promoter and instead loop to one more distal (>3/4 of detected loops!!) and a largely promotercentric connection between distinct enhancer modules. The authors leverage these tree-based networks and published single cell transcriptome profiles of the corresponding cell types to build models predicting the relative importance of each enhancer in their respective regulatory modules to cell type-specific target gene expression. They complete exciting CRISPR-based epigenome perturbation experiments in primary human islet and acinar cell types to test and validate multiple predictions. By assessing disease-associated sequence variant enrichment in and potential molecular effects on these enhancers, they nominate putative causal/functional variant-target gene relationships and uncover a general principle in which these SNPs overlap enhancers with larger predicted effects than those not containing SNPs. Moreover, they highlight potential roles for unanticipated cell types in disease susceptibility. For example, they a pancreatic ductal adenocarcinoma-associated SNP overlapping an acinar cell type enhancer and demonstrate that its CRISPR epigenetic inactivation alters XBP1 expression, implicating altered acinar cell ER stress as a potential molecular and cellular mechanism underlying genetic risk for PDAC. This study is very exciting, and the data and insights presented advance our understanding of pancreas cell type-specific transcriptional regulatory networks and key pieces of information, such as potential functional variants and their target genes, important for variantto-function translation of GWAS association signals into biological mechanisms. Pending the authors' ability to address some minor to moderate questions and concerns that arose during review, I believe this will be an exciting and impactful paper that will appeal to the readership of Cell Genomics. Questions and suggestions, divided into major and minor categories, are as follow:

Major questions/suggestions:

1) The HiChIP data as presented in the initial figures looks INCREDIBLY clean, suggesting that there are no loops at highlighted marker gene regions for each cell type in Figure 1C-H and related supplementary. Is this true that there are really no loops detected in non-target cell types? Typically, there is at least some level of looping, not necessarily specific or similar to that in the target cell type, observed in the others. Similarly, it is important to see the ATAC-seq profiles and peak calls in the other cell types. As it stands, it is impossible to objectively evaluate the quality and reproducibility of these exciting and critical profiles generated.

We appreciate the reviewer's feedback. The loci depicted in Figures 1D–H are among the most highly cell type-specific genes in our dataset. Consequently, they show robust loops only in the relevant cell type and minimal or no loops in others, underscoring the power of our cell purification and the H3K27ac HiChIP assay. To demonstrate that we do detect loops in all cell types, we broadened the genomic window in these panels to include loci with genes that are not cell type-specific (now **revised Figure 1**). Also see our response to Reviewer #1 point 2. Regarding data quality and reproducibility, we provide:

1. Raw and processed data into GEO.
2. **Revised Table S2** to include QC metrics (mapped reads, TSS enrichments, FrIP scores, valid mapped pairs) ensuring that all libraries meet or exceed current standards.
3. New correlation and replicate statistics in the supplement as requested by the reviewer (**revised Figure S1**).

Additionally, we will be releasing the data to an online portal (also see our response to Reviewer #3, point 2) that enables anyone to explore loop calls and ATAC-seq peaks across multiple genomic loci of interest. We hope these revisions offer sufficient transparency and facilitate evaluation of data quality and reproducibility by the reader.

2) Related to question 1 above, how reproducible or robust are the ATAC-seq peaks and HiChIP loops between the donor replicates for each cell type studied? It is important to provide some more targeted assessment and visualization of peak and loop calls between each biological replicate of the enriched cell types profiled. The Principal Component plots suggest clear overall distinctions between the cell types, but additional measures of correlations between replicates, such as pairwise correlation plots comparing within-cell type, between donor variation vs. between cell type variation or irreproducible discovery rate (IDR) analysis, is needed.

We thank the reviewer for the question. In addition to the principal component plots, we have now included new panels showing pairwise correlation matrices for both ATAC-seq and HiChIP loop samples, comparing each donor replicate to every other replicate (**revised Figure S1 C, E**). We observe consistently high correlation among replicates of the same cell type, distinct from across cell-type comparisons. Although HiChIP loop data tend to be noisier than ATAC-seq, we find that replicates for the same cell type cluster together in the heat map, confirming consistency in interaction frequency across anchors for each replicate pair and that most significant loops are reproducibly detected.

3) Delta cell peaks and loops are strikingly lower than other cell types despite the apparent purity and profiles shown in Figures 1 and S1. Is this related to the ultimate yield of the sorted samples being low?

Could the authors provide more details on relative yield, in addition to the compelling and impressive 95% cell purity, that they achieved? This would seem to warrant more discussion/comment in the text surrounding Figures 1, S1, or S2A. It also raises some questions about the very small delta cell enhancer trees and lack of expression specificity and transcript abundance enrichment trends observed in other cell types and highlighted in Figures 2F and Supplementary Figure 2I

We thank the reviewer for mentioning the differences observed in delta cell enhancer trees compared to other cell types. As the reviewer correctly noted, delta cells constitute only a small fraction of pancreatic islets, typically 6-8 fold lower cell yield, compared to other cell types we examined. As requested, we have added the cell yield information to the **revised Table S2**. Although our delta cell preparations consistently reached ~95% purity, the lower cell yield limited the total chromatin available for HiChIP assays. Consequently, delta cell libraries had reduced complexity, fewer number of detectable loops, and less pronounced trends of enhancer-promoter connectivity in delta cells compared to other cell types in Figure S2I. Regarding ATAC-seq data, these assays require a lot less nuclei compared to HiChIP, thus delta cell ATAC-seq profiles or the number of peaks observed are comparable to all other cell types, which is now included as a **new column in revised Table S2**. Thus, the observed difference is specifically related to chromatin input sensitivity in the HiChIP assay, not cell purity or ATAC-seq quality. Nevertheless, the delta cell loops we identified are reproducible across donors and frequent loops were observed at known loci, such as the *SST* locus (revised **Figure 1D**).

We would like to note that we omitted delta cell data from core EPIC modeling and forest analyses precisely because the fewer loops and trees introduce challenges for statistical robustness. In our original submission, we described these limitations in the Results and more explicitly in the Methods section, under EPIC modeling (copied here for clarity): “Data for d-cell was excluded due to substantially fewer number of enhancer trees detected (Figure S2A). d-cells are among the least abundant cell types within islets (less than 100,000 cells per donor). While we were able to detect frequently occurring loops, like the *SST* locus, the low cell number precludes constructing HiChIP libraries with complexity comparable to the other cell types.” As suggested, we also added a statement to the Figure S2I legend to reiterate these limitations. We hope the additional context and revisions address the reviewer’s concerns.

4) The observation that most enhancers skip their nearest linear distance gene promoter is very surprising to this reviewer. Could some of this apparent skipping be due to the certain limits in resolution to detect looping interactions between nearby (<5 kb, <10 kb) cis-regulatory elements? In essence, is there a bias of short or shorter distances for the "apparently skipped" nearest promoters? Or orientation biases driving these relative percentages? I.e., what is the relative proportion of skipped

nearest promoters on the opposite side of the potential loop (lefthand example in Figure 2D "loop-to-distal" cartoon) vs. those in which the skipped nearest promoter is within the loop?

We appreciate the reviewer's thoughtful questions. Enhancers bypassing their closest promoter have been noted by several other studies using chromatin contact assays (Miguel-Escalada et al. 2019 {PMID: 31253982}, Greenwald et al. 2019 {PMID: 31064983}, Murphy et al. 2022 {PMID: 38053013}). To address the possibility that our assays may be missing short-range interactions, we examined the linear genomic distance between each enhancer and its skipped nearest gene. Across all five cell types, the median distance to the nearest promoter was >16 kb (see table below). These results suggest that most skipped nearest promoters are not short-range outliers and are within the 5 kb detection limits of the HiChIP assays.

cell type	median distance (bp) to skipped nearest promoter
alpha	19,095
beta	19,175
delta	17,796
acinar	16,521
duct	18,711

Per reviewer's request, we also examined the position of skipped promoters relative to enhancer loops, and classified each as positioned inside or outside the loop. We found that the distribution was nearly balanced across all cell types with no consistent bias (~47% inside vs. ~53% outside). A chi-square test comparing these proportions across cell types resulted $\chi^2 = 7.21$, $df = 4$, $p = 0.125$, indicating no statistically significant differences. These results suggest that enhancer skipping is not biased by loop orientation or promoter position relative to the loop span.

cell type	% inside loop	% outside loop
alpha	47.2%	52.8%
beta	47.3%	52.7%
acinar	47.1%	52.9%
duct	47.0%	53.0%
delta	51.8%	48.2%

Together, these findings suggest that the enhancers skipping nearest promoters we and others observe is not simply a byproduct of resolution limits but may be a consequence of promoter selection. Further work will be needed to understand the mechanisms of this enhancer-promoter compatibility. We thank the reviewer for prompting this analysis and now include these results in the **revised Table S6** and **expanded the section in Methods**.

5) The tree/forest quartiles are really interesting in terms of their relative enrichment for expression specificity and increased transcript abundance. Are there unique or interesting transcription factor binding motifs (TFBMs) or other sequence features that distinguish or are enriched in the large vs. small tree enhancer "forests"? In essence, are there distinct transcription factors or general DNA binding factors that nucleate or stabilize large vs. small enhancer modules identified?

We thank the reviewer for this insightful question. To clarify, our analysis in Figure 2F examined the correlation between enhancer tree size (number of enhancer nodes per gene) and gene expression specificity or

abundance, rather than enhancer forests spanning multiple genes. Regardless, we performed a TF motif enrichment analysis comparing enhancer sequences from the largest 25% of enhancer trees to those from the smallest 25%. Indeed, we found significant enrichment of lineage-specific transcription factor motifs specifically in the largest enhancer trees, whereas smaller trees did not yield any TF motif that passed the significance threshold (**Author Response Figure, RF1**). These findings align with the idea that large enhancer clusters often contain lineage-defining TFs, possibly supporting robust expression of key cell identity genes. We think that fully addressing how these transcription factor motifs influence enhancer cluster formation would require additional functional experiments beyond our current scope. However, we agree with the reviewer that this finding is interesting and worthy of future investigation.

6) Approximate correlations in enhancer forest and TAD genomic span sizes is intriguing. Can the authors take this a step further and compare specific enhancer forest spans to their corresponding TADs in the genome? I.e. do the enhancer forests overlap TADs in each of the chromosome regions? Or is there no clear correlation between enhancer forest spans and TADs per location?

We appreciate the reviewer's interest in the relationship between enhancer forests and TAD structures. To explore this, we obtained a pancreas Hi-C dataset from the

ENCODE Consortium (GEO accession number GSE237250) and used contact domain calls generated by the Arrowhead algorithm (Durand et al PMID: 27467249). Similar to our findings in response to Reviewer 1 (point #5) regarding enhancer hubs, we observed that nearly all TADs were contained within our enhancer forests, while the forests generally span larger genomic intervals than individual TADs. The median Jaccard index between enhancer forests and TADs was 0.5 (median overlapping forest span 1.3 Mb, median overlapping TAD span 0.9 Mb).

However, as the reviewer likely anticipates, such comparisons between forests and TADs are not straightforward. Benchmarking studies (Dali et al. 2017, PMID: 28334773; Xu et al., 2024, PMID: 38782890) have shown that TAD boundaries and domain definitions vary substantially across computational tools and are sensitive to resolution, particularly when accounting for the hierarchical nature of 3D genome organization (e.g., subTADs, metaTADs). Since enhancer forests could align more closely with different layers of TAD hierarchy depending on the locus, a comprehensive multi-layer comparison would indeed be necessary to properly address the reviewer's insightful question.

We think the current level of comparison supports the general observation of partial overlap but is insufficient to draw more specific conclusions. We therefore prefer to avoid over-interpreting these results in the manuscript and **have removed** the TAD-related statement from the main text.

7) Are the apparent cell type-specific tree structures and connections unique to the cell type vs. its related endocrine or exocrine counterparts? I.e., alpha-specific vs. all endocrine or ductal cell vs. all exocrine? Or is this a one-vs-all comparison? Are these apparent specificity features driven by the strong out-group comparisons (for example beta vs. acinar or ductal) rather than the similar cell types (beta vs. alpha, delta)? Are there interesting themes and insights that emerge from 'endocrine' vs. 'exocrine' comparisons? Or 'within-compartment' comparisons? We thank the reviewer for highlighting a point that warrants clarification regarding our terminology. In our manuscript, the term "cell type-specific enhancer-promoter trees" refers to enhancer-promoter interactions identified within a given purified cell type population as opposed to those identified from bulk pancreas tissue. This term reflects the cell type-resolved nature of the data, not differential interactions belonging to a single cell type. For example, a-cell trees can contain enhancers common across multiple cell types (like *ACTIN* locus enhancers), enhancers shared within compartments (like *CHGA* for endocrine), and enhancers uniquely detected in a-cells (like the *GCG* locus).

Importantly, our dataset provides sufficient information to enable straightforward extraction and analysis of enhancer interactions at different levels, whether unique to a specific cell type or shared within a cellular compartment, depending on the specific biological question. **We have now added the following clarification in the Methods:**

“Throughout the manuscript, we use the term ‘cell type-specific enhancer-promoter trees’ to describe enhancer-promoter interactions detected specifically within purified cell populations. This designation emphasizes the cell type-resolved nature of the data and does not necessarily imply differential interactions. Consequently, enhancer trees defined for one cell type may contain interactions unique to that cell type, interactions shared among related cell types, or interactions common across multiple pancreatic cell populations.”

Further, we agree with the reviewer’s intuition that specificity observed in gene expression patterns could be partly driven by broader compartment level and cell typespecific differences. The *PCSK2* locus illustrates exactly this scenario (the revised Figure S5H). Quantitatively, the *PCSK2* enhancer tree in a-cells contains 59 enhancer nodes, while the b-cell tree contains only 24 nodes. Approximately 20 nodes are shared between a- and b-cells, reflecting common regulatory interactions. However, the remaining enhancer nodes are unique to a-cells (upstream of the gene body), likely include regulatory elements driving higher *PCSK2* expression. In our CRISPR experiments, targeting these a-cell-specific loci increased *PCSK2* transcription in b-cells, indicating that the elements can modulate *PCSK2* levels. Examining these differential interactions systematically is beyond the scope of the current manuscript but represents an important future direction.

8) Related to 7 above, this reviewer is very curious if these distinctive enhancer modules or forests reveal differential gene set or pathway enrichments? What about motifs? Do these distinctive modules reveal key (and different) transcription factors that might nucleate or stabilize these distinct enhancer forests?

We appreciate the reviewer’s question. As noted in our response to reviewer’s point #5, given the large genomic span and multi-gene nature of forests, defining appropriate backgrounds and controls for forest-level motif or pathway enrichment analysis is not straightforward.

RF2. GO Term analysis on large and small enhancer trees

However, to address the reviewer's interest in functional and regulatory distinctions between larger and smaller enhancer modules, we performed Gene Ontology (GO) enrichment analysis comparing genes associated with the largest 25% versus smallest 25% of enhancer trees within each cell type. The results (**Author Response Figure RF2**) reveal distinct pathway enrichments between large and small enhancer trees in each cell type. For example, in b-cells, genes with large enhancer trees were enriched

for pathways related to hormone secretion and insulin secretion, while small-tree genes showed no significant GO term enrichment. Similar cell type-relevant patterns emerged for acinar, duct, and α -cells. We recognize that these analyses do not fully resolve the biological mechanisms driving forest-scale organization, but we hope the new tree-level GO and motif enrichment results provide useful insight into the regulatory features distinguishing large versus small enhancer modules.

9) It was surprising and not particularly intuitive to read that the linear and full tree α -cell models performed similarly and suggested a distinct organization of α -cell specific genes in the genome. Can the authors expand upon this observation and assertion and comment or speculate on why this might be the case and what this might mean? Is this apparent distinct α -cell specific gene organization reflected in other data collected, such as the number of peaks or the proximity or density of α -cell interactions vs. the other cell types?

We thank the reviewer for this thoughtful comment. We agree that the similar performance between the linear and full tree α -cell models was surprising. We speculate that α -cell-specific genes may be in proximity to each other in the human genome because of genome evolution, without interspersed non α -cell genes, hence allowing linear enhancer-promoter proximity to estimate regulatory interactions nearly as effective as our physical maps (see Figure 1B, *GCG*, *FAP*, *DPP4*, all α -cell-specific genes positioned in proximity). However, this hypothesis would require additional analysis to confirm and is not central to the main findings of the manuscript. We have **removed these statements** from the text, and subordinated the model comparisons to the supplement (now Figure S4) to improve clarity and keep the focus on EPIC's enhancer prioritization across cell types.

10) CRISPR enhancer targeting in Figure 5 is innovative and exciting! Can the authors also show in the main or supplementary figures the per-donor targeting effects? Since human islets can be variable in terms of behavior, quality, etc., it will take the robustness and reproducibility of these observations even further by seeing it is consistent between the two or more independent donors tested according to the figure legend. Additionally, it would be informative to include representative images of the promoter-targeting gRNA experiments alongside the selected enhancers in Figures 5E and 5H.

We thank the reviewer for highlighting the importance of donor-to-donor reproducibility in the CRISPR enhancer targeting experiments. We now include **additional data** from independent donors not shown in the original submission (**revised Figures S5 and Figure 6**) addressing these points (also the reviewer's related minor point #5). These new graphs show consistent CRISPR perturbation effects at the *PCSK1*, *PCSK2* and *XBP1* enhancers across donors. We note that due to the limited availability and quality of human islet tissue, it was not always possible to test all enhancer combinations from a given locus in every donor. In many cases, subsets of enhancers were prioritized based on cell yield. The main and supplementary figures focus on experiments where multiple enhancers at the same locus were tested in parallel within the same donor. To improve transparency, we also now provide **the full raw RNA-FISH probe intensity data from all donors in table format (revised Table S9)**. As requested, we have also **updated the main Figures 5 and 6** to include representative images from promoter-targeting gRNA experiments. We think these additions further strengthen the rigor and reproducibility of the functional experiments and thank the reviewer for prompting these clarifications.

11) Where do the enhancers to the right of *PCSK2* in Figure S4D rank? I.e., are those predicted to be stronger enhancers but just not specific? Or are they poor-ranking candidates?

We provide Author Response Figure RF3 showing the distribution of EPIC effect sizes for enhancers located downstream of the *PCSK2* transcription start site (black dots). These downstream enhancers are shared between a- and bcells and exhibit a wide range of EPIC scores. Because our focus was on cell type-specific regulation, our CRISPR experiments targeted enhancers that looped to the *PCSK2* promoter only in a cells. As noted in our response to point 7, we agree that a systematic analysis of

shared versus cell type-specific enhancer-promoter interactions would be a valuable direction for future work.

12) It was surprising that enhancer targeting with the strong VP64 activator was more powerful than promoter targeting. Why do the authors think this is the case? Is it possible these strong enhancers may serve as other promoters? Or are they very clear intragenic or intergenic enhancers? Are there other

features of these enhancers that might predict their stronger regulatory effects? Sequence motifs? High eRNA levels vs. low?

We agree with the reviewer that stronger activation from enhancer targeting compared to promoter targeting was intriguing. One plausible explanation is that the chromatin context and optimal positioning of sgRNAs might influence CRISPR-VP64 activity more than whether the target is a promoter or enhancer. Prior work from the Weissman lab showed that CRISPRa is highly sensitive to gRNA positioning around the TSS and nucleosome occupancy (Horlbeck et al 2016, PMID: 26987018). The enhancers we targeted were selected based on strong EPIC scores and chromatin features, thus making them more permissive to coactivator recruitment upon VP64 targeting.

A systematic analysis of enhancer features (sequence motifs, enhancer RNA levels) that might predict CRISPR activation is beyond our current scope, but we agree this would be informative in future work.

13) Related to 12 above, why do the authors think similar targeting didn't activate expression in exocrine cell types? Do chromatin states or ATAC-seq maps give a clue? Why does targeting the gene promoter with strong VP64 transactivator not even elevate gene expression in these cell types?

We appreciate the reviewer's thoughtful questions. As noted above, we consider the chromatin context, nucleosome positioning, and specific sequence features critical determinants of dCas9-based activation efficiency. Our chromatin data (**revised Figure S5 G-H**) clearly demonstrate significant differences in chromatin context between endocrine and exocrine cells at the targeted loci. Further, *PCSK1* and *PCSK2* promoters may be actively suppressed in exocrine cells, which may hinder efficient recruitment of transcriptional coactivators, like the Mediator complex. Thus, even strong recruitment of VP64 may fail to drive significant transcription in a non-permissive chromatin context lacking relevant co-activators or transcription factors.

14) The observation that SNP-containing enhancers have higher predicted EPIC effect sizes was VERY interesting. While I appreciate the focus on unexpected phenotypes and cell types for the GWAS enrichment studies, it seems really important to include the type 1 and type 2 diabetes analyses and interpretations before switching to acinar and cancer connections. Given the extensive focus on endocrine cell type features and analyses, this switch seemed abrupt and not well-justified. I strongly encourage further inclusion and work-up of diabetes and related phenotype results with the endocrine cell type maps and models in the main and supplementary figures. This seems to be an important missed opportunity to expand our knowledge and understanding of diabetes GWAS target genes and

potential regulatory features and insights. We appreciate the reviewer's enthusiasm and agree that our findings linking diabetes-related GWAS variants to islet cell regulatory networks are valuable. The focus on pancreas cancer was primarily guided by funding priorities, thus limiting our scope for extensive experimental validation in diabetes-related phenotypes. Nonetheless, we have expanded our EPIC-based GWAS risk SNP analysis and examined three pancreatic traits (type 1 diabetes, type 2 diabetes, and pancreas cancer) across three cell types

(acinar, duct, b-cells; also see our response to Reviewer #1 point 6). These analyses encompass more than 7,500 enhancers and 300 candidate genes, providing a substantial resource for the diabetes research community (**revised Table S10**). We hope the detailed datasets and analyses will facilitate further investigation by the diabetes researchers.

15) In Supplementary Figure 5A-C, it would be helpful to show the corresponding data for the other cell types so one can compare their enhancer predictions and maps with their respective S5C expression.

We thank the reviewer for the suggestion. We are providing **a new supplementary figure (Figure S6)** to display the *GATA4* data for the other cell types.

16) The switch from CRISPRa assessment for Figure 5 *PCSK* enhancer experiments to CRISPRi for Figure 6 acinar enhancer experiments are not well justified. Was there a particular reason to apply consistent perturbations between these experiments? Or to show for one or both complementary effects of CRISPR activation vs. inhibition? Was CRISPRi selected because the PDAC risk allele is predicted to decrease enhancer accessibility, activity, or target gene expression?

We appreciate the reviewer's question and take this opportunity to clarify our rationale. In the original submission, we noted that "Because there is no detectable *PCSK1* expression in a- or exocrine cells, we opted to use the CRISPRa system (VP64) to examine cell type-specific effects of *PCSK1* enhancer perturbations." In other words, our goal was to determine whether these enhancers could specifically activate transcription in cells where the baseline expression was effectively zero. CRISPRi was not suitable in this context since knocking down expression is impossible if baseline transcription is absent. The rationale for using CRISPRa for *PCSK2* locus was similar— no baseline expression in exocrine cells.

For the *XBP1* locus in Figure 6, because *XBP1* is expressed across all tested cell types, we initially intended to test enhancer function using both CRISPRa and CRISPRi, to assess whether enhancers with strong repressive effects also showed strong activating potential. However, we encountered technical difficulties regarding gRNA design (as detailed in points 12 and 13). gRNAs optimized for CRISPRi did not yield effective activation with CRISPRa, and vice versa.

Given the high cost and complexity of designing and validating separate gRNA libraries for both systems, we chose to proceed with the CRISPRi approach, reasoning that *XBP1* was already robustly expressed in all cells and CRISPRa might encounter an "upper limit" in driving further activation.

17) The authors' implication of *XBP1* as an acinar cell-specific expressed gene is odd given well-documented roles in beta-cell and islet protein folding and endoplasmic reticulum stress. Also, it is unclear how or why Figure 5E suggests it has a higher expression specificity score in acinar cells although it appears expressed (at comparable levels) in alpha and beta cells. Is the acinar cell specificity of this variant and *XBP1* regulation better supported by comparing and contrasting the regulatory profiles and enhancer forests in acinar vs. the other cell types profiled? Can this be shown in main or supplemental figures to better document and motivate the specific selection of acinar cells as the cell type of interest and importance? This should also apply to Supplementary Figure 5 so one can assess the relative structures and specificity of *GATA4* trees, profiles, and loops.

We thank the reviewer for this comment and appreciate the opportunity to clarify. Indeed, as the reviewer correctly notes, *XBP1* is robustly expressed in several pancreas cell types and known to have critical roles in b-cell function and ER stress. However, our previous analysis integrating multiple pancreas single-cell RNA-seq datasets (Sturgill et al., PMID: 38238687) highlighted significantly higher expression of *XBP1* specifically in acinar cells compared to other cell types. Below, we are showing the normalized transcript counts (TPM) of *XBP1* from these single cell studies:

acinar	alpha	beta	delta	duct
59221.4	14561.7	16383.6	18006.8	11139.5

Thus, while expressed across multiple cell types, *XBP1* expression is nearly 6-fold higher in acinar cells compared to ducts, and about 4-fold higher than endocrine cells. Acinar cells synthesize, store and secrete vast amounts of peptides (estimated 70% of their total protein content), likely necessitating higher *XBP1* expression to manage the burden on ER stress. Nonetheless, we recognize the ESS panel did not effectively reflect these nuances, **we have now removed it from the revised manuscript.**

To further address the reviewer's request for clarity, we now include a UCSC browser view comparing chromatin interactions at the *XBP1* locus across all profiled cell types in a **new supplementary figure (Figure S8)**. In line with its expression, all cell types show detectable chromatin interactions at the *XBP1* locus, with acinar cells showing the highest density and complexity of loops (**Figure S8A**). In addition, we provide the *XBP1* enhancer tree network

models for all cell types (**Figure S8B-E**). These models, plotted on a linear scale to highlight differences clearly, emphasize the distinct chromatin profiles among cell types.

The selection of *XBP1* enhancers for functional perturbations was informed by multiple converging lines of evidence. First, the GARFIELD analysis found that acinar cell specific enhancers were significantly enriched for PDAC-associated GWAS variants (Figure 6A, **revised Table S10**). Second, our enhancer tree analysis linked the enhancers overlapping with PDAC-associated variants to the *XBP1* promoter in acinar cells. Third, EPIC predicted that these SNP-overlapping enhancers would have an impact on *XBP1* expression when perturbed. However, as noted by the reviewer and evident in the new Figure S8, these enhancer-promoter interactions are not exclusive to acinar cells, as some of the enhancers also loop to the *XBP1* promoter in other cell types, and it is possible that they might indeed regulate *XBP1* transcription in other cell types.

We hope these clarifications and additional data strengthen our rationale and address the reviewer's valuable point.

18) In the Discussion second-to-last paragraph, perhaps the authors could include consideration and discussion of the potential for monogenic disease mutations or other rare variants to be present and identified.

We appreciate the reviewer's suggestion to consider rare or monogenic disease variants. While the primary focus of our current study is common disease-associated variants identified via GWAS, we fully agree that enhancer annotations and cell typespecific regulatory maps generated in our study provide a valuable resource for interpreting rare, potentially pathogenic variants. For example, Wakeling et al. 2022 (PMID: 36333503), Bennett et al. 2025 (PMID: 40033430) identified rare noncoding variants within a regulatory element controlling the hexokinase gene (*HK1*), causing congenital hyperinsulinism through aberrant gene expression in pancreatic b-cells. Similar to our findings, these examples highlight the critical role enhancers play in regulating key cell type-specific gene expression, and underscore the potential of enhancer mapping and predictive models like EPIC to inform studies on rare diseasecausing variants. We have now **expanded our discussion** to acknowledge this important point as follows:

"While our analyses focused on common genetic variants associated with complex pancreatic traits, it is important to consider that rare, high-impact germline variants or mutations within enhancers similarly disrupt gene expression and lead to monogenic disorders. For instance, rare noncoding variants within an enhancer of hexokinase gene (*HK1*) were recently shown to cause congenital hyperinsulinism by mis-regulating expression in pancreatic b-cells (Wakeling et al., 2022, Bennett et al., 2025). Our enhancer maps and the EPIC prioritization framework could

thus greatly facilitate the localization and mechanistic characterization of such variants, enabling deeper insights into the genetic basis of rare monogenic diseases."

Minor questions/revisions:

1) Text is confusing as to how many donors/samples have been profiled. In the abstract it indicates 20 but end of Intro paragraph indicates 27. It would be helpful if the authors could be more transparent, clear, and consistent as to how many individuals' profiles the data represent.

We apologize for the confusion and have **revised supplementary Table S1** to clearly list all donor IDs, demographic information, and the assays performed. 28 unique donors were used for chromatin profiling (ATAC-seq and HiChIP), and 13 additional donors were used for CRISPR perturbation experiments, bringing the total to 41 donors represented in the study. In some cases, acinar and islet tissues from the same donor were assigned different RRIDs by the organ procurement centers, which may have contributed to the inconsistency. We have also uploaded the full donor-assay lists to GEO and are now including **a new supplementary table** to provide **the full raw probe intensity data from all donors in CRISPR/HCR experiments (Table S9)**. We thank the reviewer for this comment and hope these updates provide sufficient clarity.

2) Figure 2E and other relevant expression specificity score legends: define "expression specificity score" prior to the abbreviation ESS. It appears the "score" or other term for the second "S" in ESS is missing from the definitions in the legends.

We thank the reviewer for mentioning this inconsistency. We have edited the legends.

3) Supplementary Figure 2G: "loop-skp gene pairs" should be changed to "loop-skip gene pairs"

We have made the corrections.

4) Page 7, change "Supplemental Figure 2B-D" to "Supplementary..." to be consistent with other references to supplementary figures. This appears to be an occasional recurrent classification difference that should be checked and harmonized throughout the text.

We thank the reviewer for pointing this out. We have updated the figure calls based on Cell Genomics author guidelines.

5) Figure 5B and related transduction efficiency evaluations: can the authors provide FACS or other more quantitative measures and metrics? It is entirely possible that alpha and beta cells could exhibit differences in transduction efficiency.

We appreciate the reviewer's important suggestion. In the **revised supplemental figure S5A-D**, we now provide the total number and percentage of mCherry⁺ cells across all donors used in the study, broken down by cell type for each adenoviral transduction. This analysis directly addresses the concern regarding potential differences in viral transduction efficiency across cell types and donors. As the reviewer anticipated, we did observe cell type-specific differences, with a-cells showing the lowest transduction efficiency. Nonetheless, these differences do not impact our quantification or conclusions. For every gRNA condition, we only considered cells that were both mCherry and relevant cell type marker positive. Cells that were not transduced were excluded from the analysis. As noted in the figure legend, each condition included hundreds to thousands of double-positive cells. We thank the reviewer for prompting this additional analysis, which helped us clarify the extent of variability in viral transduction while reinforcing the robustness of our experimental design.

6) Figures 5F, 5I: y-axis values appear to be missing for alpha cells. Is it the same as betas? If so, the authors should plot them all together on one large scatter plot. If not, it is important to include the different y-axis values so one can compare the relative magnitude of effects these epigenetic perturbations elicit.

We apologize for this oversight and corrected the y-axis labels accordingly.

7) Discussion, page 18:

- a. "that most enhancers" phrase is duplicated. One instance should be deleted
- b. Change "potentially an additive model" to "potentially additive, model"
- c. Suggest to change "potentially, already positioned for transcription" to "potentially prepositioned for transcription..."
- d. Change "like HiChIP is they capture chromatin" to "like HiChIP is that they capture..."
- e. Remove comma following references 73,74

We thank the reviewer for these detailed suggestions. We have made the recommended corrections to the text. Regarding the comma following references 73 and 74, it reflects the formatting required by the journal's reference style, and we will defer to the publisher's formatting conventions.

Reviewer #3

Wang et al reported the development of a new strategy to prioritize enhancers in cells related to islets and their association with pancreas disease risk. Authors performed H3K27ac HiChIP and ATAC-seq experiments in cells sorted from pancreas organ donors. They then build a graphbased network for enhancer-enhancer interactions. They studied the properties of the tree across different cell types. They then used machine learning to predict how perturbation in the enhancer network controls gene

expression. Finally, they used RNA FISH and validated predictions of their machine learning using CRISPRa and CRISPRi, adding or removing enhancer activities.

This study is very rigorous, cell types are purified from organ donors and experimental set up is well suited for studying enhancer-enhancer network analysis. The computational approach to decipher the complexity of enhancer interaction is innovative and well justified. Validation data using RNA FISH increases the confidence for machine learning predictions. I have two questions for the authors:

1) There is an opportunity to define the nature of genes with distinct enhancer topology across different cell types. For example what are the specific genes with high level of enhancer connectivity? do transcription factors, genes related to metabolism or other categories have high connectivity? What is the nature of genes with low connectivity? Are these patterns different across cell types?

We thank Reviewer #3 for this thoughtful question. Similar points were raised by Reviewer #2 (see our responses to Reviewer #2, points 5 and 8), and we summarize those findings here while including additional specifics in response to your comment. As the reviewer notes, our analysis of enhancer-promoter topology examined how enhancer tree size (i.e., the number of enhancer nodes per gene) relates to expression specificity and transcript abundance (Figure 2F). To further explore the types of genes associated with different levels of enhancer connectivity, we performed Gene Ontology (GO) enrichment analyses comparing genes in the top 25% versus bottom 25% of enhancer tree size within each cell type (**Author Response Figure RF2**). These results show that genes with large enhancer trees are enriched for lineage-relevant functions— hormone secretion in b-cells, digestive enzyme production in acinar cells. In contrast, genes with smaller enhancer trees generally lacked significant GO term enrichment, suggesting they may include more ubiquitously expressed or housekeeping genes.

To directly address the reviewer's question about the "nature" of genes with distinct enhancer topology, we list below the top 50 genes within the large enhancer trees in acinar cells as an example:

GDF10, CLDN10-AS1, LINC01799, BANF2, LINC01625, **CELA2A**, **CPA1**, **AQP8**, **CELA3A**, **IL22RA1**, **PNLIPRP1**, **CTRB1**, **CTRB2**, **CELA3B**, **CPA2**, **CELA2B**, **PNLIP**, **PNLIPRP2**, **PLA2G1B**, **PDIA2**, **LINC02185**, **MYBL2**, **TMEM266**, **CLPS**, **CTRL**, **CPB1**, **CTRC**, **LINC01624**, **GATA4**, **BCL2L14**, **GAS1**, **BCAT1**, **LINC00940**, **HEYL**, **PRSS3**, **FGL1**, **PLAT**, **CCRL2**, **SNORD17**, **SNTB1**, **MECOM**, **NIBAN1**, **HHLA2**, **CPA4**, **SNORA63**, **PODXL**, **GATA2-AS1**, **ITGAX**, **CXCL12**, **KCTD15**

These genes encompass a diverse set of molecular functions (highlighted in bold), including secreted enzymes and peptides, transcription factors, and long non-coding RNAs. This diversity suggests that large enhancer trees are not limited to one functional class. Rather, these enhancer architectures likely reflect the need for robust and tightly controlled expression of

genes critical for cell identity and function. These patterns are consistent across the pancreatic cell types analyzed and reinforce the conclusion that enhancer tree size correlates with regulatory and functional specialization. We thank the reviewer for prompting this clarification.

2) I recommend authors to create a web portal and enable the community to browse through the enhancer network connectivity across different cell types. This type of web portal can have a broad impact enabling pancreas researchers to test different hypotheses using this valuable resource.

We thank the reviewer for this recommendation and fully agree that publicly accessible web portal would greatly enhance the utility of our datasets. We are pleased to share that NIH has funded the Pancreas Knowledge Base (PanKbase) initiative (<https://pankbase.org/>)— a centralized, open-access resource for curated datasets, computational tools, and workflows supporting pancreas research. We plan to deposit our data into PanKbase to facilitate broad community access. In addition, we will upload the datasets to the UCSC Genome Browser and provide a public track hub link for interactive browsing.

Referees' reports, second round of review

Reviewer #1: The authors adressed all my comments and in my opinion the manuscript is now suitable for publication

Reviewer #2: The authors took a thoughtful, measured, and reasonable approach to address my comments and answer my questions. The updated edits to the text and figures accordingly are appreciated. I have no further comments, questions, or concerns.

Congratulations on an exciting study that is well-designed and well-executed—I anticipate this will be an important contribution to the field and cited as such!

Reviewer #3: The authors addressed my previous concerns and I don't have additional concerns.